# $AutoDrive$-$P^3$: Unified Chain of Perception–Prediction–Planning Thought via Reinforcement Fine-Tuning

**Yuqi Ye[1,*], Zijian Zhang[1,*], Junhong Lin[1], Shangkun Sun[1], Changhao Peng[1], Wei Gao[1,2,†]**
[1]School of Electronic and Computer Engineering, Peking University
[2]Guangdong Provincial Key Laboratory of Ultra High Definition Immersive Media Technology

## Abstract

Vision-language models (VLMs) are increasingly being adopted for end-to-end autonomous driving systems due to their exceptional performance in handling long-tail scenarios. However, current VLM-based approaches suffer from two major limitations: 1) Some VLMs directly output planning results without chain-of-thought (CoT) reasoning, bypassing crucial perception and prediction stages which creates a significant domain gap and compromises decision-making capability; 2) Other VLMs can generate outputs for perception, prediction, and planning tasks but employ a fragmented decision-making approach where these modules operate separately, leading to a significant lack of synergy that undermines true planning performance. To address these limitations, we propose $AutoDrive$-$P^3$, a novel framework that seamlessly integrates **P**erception, **P**rediction, and **P**lanning through structured reasoning. We introduce the $P^3$-$CoT$ dataset to facilitate coherent reasoning and propose $P^3$-$GRPO$, a hierarchical reinforcement learning algorithm that provides progressive supervision across all three tasks. Specifically, $AutoDrive$-$P^3$ progressively generates CoT reasoning and answers for perception, prediction, and planning, where perception provides essential information for subsequent prediction and planning, while both perception and prediction collectively contribute to the final planning decisions, enabling safer and more interpretable autonomous driving. Additionally, to balance inference efficiency with performance, we introduce dual thinking modes: detailed thinking and fast thinking. Extensive experiments on both open-loop (nuScenes) and closed-loop (NAVSIMv1/v2) benchmarks demonstrate that our approach achieves state-of-the-art performance in planning tasks. Code is available at https://github.com/haha-yuki-haha/AutoDrive-P3.

## 1 Introduction

Autonomous driving aims to predict trajectories that are both comfortable and collision-free by leveraging environmental and ego-vehicle information. Traditional approaches decouple the autonomous driving pipeline into three independent stages: perception (Li et al., 2024c; Liang et al., 2022), prediction (Zhou et al., 2023; Shi et al., 2024), and planning (Huang et al., 2024b; Liu et al., 2025). However, these module design often leads to error accumulation, which significantly degrades the final trajectory quality. Recent years have witnessed significant advancements in end-to-end training for autonomous systems (Hu et al., 2023; 2022; Jiang et al., 2023), as shown in Fig. 1(a). Nevertheless, these small-scale end-to-end models are constrained by limited dataset size and model capacity, resulting in a lack of world knowledge and poor performance in long-tail scenarios.

To address long-tail scenarios, recent works (Tian et al., 2024; Wang et al., 2024; Zhou et al., 2025a;b; Yuan et al., 2025; Bai et al., 2026) introduce Vision-Language Models (VLMs) into autonomous driving. Leveraging large-scale pre-training, VLMs show strong adaptability to diverse scenarios. However, current VLM-based end-to-end systems face three key limitations: **1) Lack of**

---

*Core contributor. Please see appendix for individual contributions. † Corresponding author.

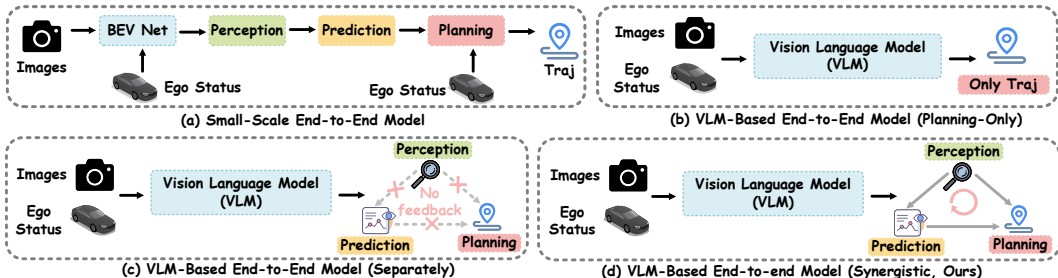

Figure 1: **The difference between** $AutoDrive$-$P^3$ **and other paradigms.** Our method combines an end-to-end training framework with a three-stage collaborative supervision form with VLM.

**Chain-of-Thought (CoT) supervision**: VLM-based systems benefit from CoT, but some VLMs directly output trajectories (Fig. 1(b)), limiting reasoning for decision-making. **2) Lack of multi-task synergy**: Although most VLMs (Zhou et al., 2025a; Wang et al., 2024) can answer perception, prediction, and planning queries (Fig. 1(c)), they treat these tasks separately, resulting in poor synergy and weak planning. **3) Planning-only GRPO supervision**: Existing Group Relative Policy Optimization (GRPO) applications optimize only planning metrics such as L2 distance or closed-loop performance (Zhou et al., 2025b; Yuan et al., 2025), leaving perception and prediction without direct supervision. This yields superficial gains, limited interpretability, and unreliable planning.

We argue that these limitations lies in the failure to capture the staged CoT process across perception, prediction, and planning. Autonomous driving fundamentally requires these three stages to work in synergy, where accurate perception enables reliable prediction, and both are indispensable foundations for robust planning. However, conventional approaches with planning-only optimization neglect this interdependence, treating perception and prediction as byproducts rather than core components. Accordingly, we reconsider the role of GRPO in autonomous driving. Rather than restricting supervision to the planning stage alone, GRPO should be extended to explicitly encompass perception, prediction, and planning within a unified chain. Such a formulation ensures synergistic interactions across three modules and promotes coherent reasoning throughout the entire pipeline.

To address above fundamental limitations, we propose a novel three-module supervised GRPO algorithm specifically designed for the $AutoDrive$-$P^3$ framework, as illustrated in Fig. 1(d), which unifies **P**erception, **P**rediction, and **P**lanning into a cohesive architecture. The $AutoDrive$-$P^3$ framework is capable of not only answering perception and prediction queries but also enhancing planning performance through synergistic interactions among all three modules. During the Supervised Fine-Tuning (SFT) stage, we train the model using our proposed $P^3$-$CoT$ dataset, resulting in the $AutoDrive$-$P^3$ base model. This model can generate responses following a structured perception-prediction-planning CoT format, thereby reducing the domain gap between VLMs and autonomous driving systems. Subsequently, inspired by the GRPO algorithm (Shao et al., 2024; Guo et al., 2025; Zhang et al., 2025), we propose $P^3$-$GRPO$ algorithm, which is a novel hierarchical and progressive optimization reinforcement fine-tuning (RFT) method that provides explicit supervision across perception, prediction, and planning modules. The $P^3$-$GRPO$ algorithm not only improves the accuracy of perception and prediction but also significantly enhances the model's planning capability by ensuring coherent and context-aware decision-making.

We extensively evaluate $AutoDrive$-$P^3$ using real-world datasets, including the closed-loop NAVSIMv1/v2 (Dauner et al., 2024; Cao et al., 2025) and the open-loop nuScenes (Caesar et al., 2020). Experimental results demonstrate that $AutoDrive$-$P^3$ achieves superior performance across various end-to-end autonomous driving benchmarks under both open-loop and closed-loop settings. More importantly, experimental results validate that our proposed $P^3$-$GRPO$ algorithm significantly enhances planning performance through its hierarchical and progressive supervision mechanism, which systematically improves perception and prediction capabilities and consequently leads to more reliable and accurate planning decisions. Additionally, to balance inference efficiency with performance, we introduce dual thinking modes: detailed thinking and fast thinking. The main contributions of this paper are summarized as follows:

1. We present $AutoDrive$-$P^3$, an end-to-end vision-language driving framework that resolves a key limitation of current VLMs by explicitly capturing the relationship between perception, prediction, and planning in autonomous driving.

2. We introduce a three-module supervised $P^3$-$GRPO$ algorithm that provides hierarchical and progressive optimization across perception, prediction, and planning tasks, significantly enhancing reasoning coherence and planning reliability by our proposed $P^3$-$CoT$ dataset. Additionally, to balance efficiency with performance, we introduce dual thinking modes: detailed thinking and fast thinking.

3. We demonstrate that $AutoDrive$-$P^3$ achieves state-of-the-art performance on multiple autonomous driving benchmarks, including both open-loop and closed-loop tests, underscoring the effectiveness and generality of our approach.

## 2 RELATED WORK

### 2.1 END-TO-END AUTONOMOUS DRIVING METHODS

Autonomous driving systems have transitioned from traditional modular designs—featuring decoupled perception, prediction, and planning modules—toward end-to-end learning frameworks. Representative methods such as UniAD (Hu et al., 2023), ST-P3 (Hu et al., 2022) and VAD (Jiang et al., 2023) integrate these tasks into a single model trained jointly, improving planning performance. DiffusionDrive (Liao et al., 2025) integrates diffusion into trajectory planning, and WoTE (Li et al., 2025) leverages a BEV-based world model to predict future agent states, enabling online trajectory evaluation and selection. Though end-to-end autonomous driving methods make great progress, they still suffer from a lack of world knowledge and poor performance in long-tail scenarios.

Due to the limited capacity of such compact models and their constrained semantic understanding of complex environments, recent efforts increasingly incorporate Vision Language Models (VLMs) into driving systems. Approaches including DriveVLM (Tian et al., 2024), EMMA (Hwang et al., 2024), VLM-AD (Xu et al., 2024), OpenEMMA (Xing et al., 2025), OmniDrive (Wang et al., 2024), OpenDriveVLA (Zhou et al., 2025a), and AutoVLA (Zhou et al., 2025b) benefit from VLMs' rich world knowledge and reasoning capabilities, demonstrating strong performance in driving scenarios. Nonetheless, while these methods are capable of answering QA-style queries about perception, prediction, and planning, they often address each task in a fragmented manner rather than through unified modeling. This lack of integration prevents the planning module from fully leveraging perceptual and predictive features, ultimately limiting overall planning performance.

### 2.2 GROUP RELATIVE POLICY OPTIMIZATION

The Group Relative Policy Optimization (GRPO) algorithm (Shao et al., 2024; Guo et al., 2025), introduced by DeepSeek, has demonstrated strong potential in enhancing the reasoning capabilities of Large Language Models (LLMs). With Vision-R1 (Huang et al., 2025) applying GRPO to Vision-Language Models (VLMs) and R1-VL (Zhang et al., 2025) further adopting step-wise reward mechanisms, GRPO has proven effective in improving VLM-based reasoning. In the context of autonomous driving, several works, such as AutoVLA (Zhou et al., 2025b), Plan-R1 (Tang et al., 2025), AlphaDrive (Jiang et al., 2025), and AutoDrive-R² (Yuan et al., 2025), have successfully incorporated GRPO to enhance the performance of driving-oriented VLMs. While these methods achieve notable results, they primarily rely on supervised learning only on the final planning outputs, without reinforcing perception and prediction modules through reward guidance. This narrow focus limits the synergistic effects between reasoning and low-level control, thus constraining the full potential of integrated planning capabilities.

## 3 PRELIMINARIES

**VLM-based End-to-end Autonomous Driving Problem Formulation.** We model end-to-end autonomous driving as mapping inputs to a trajectory $Traj = \{(x_t, y_t)\}_{t=0}^T$, where $(x_t, y_t)$ is the ego vehicle's position at time $t$. Given ego state $E$, sensor data $S$, and commands $C$, the trajectory distribution is autoregressively factorized as:

$$P(Traj \mid E, S, C) = \prod_{t=0}^{T} P\big((x_t, y_t) \mid E, S, C, (x_0, y_0), \ldots, (x_{t-1}, y_{t-1})\big). \qquad (1)$$

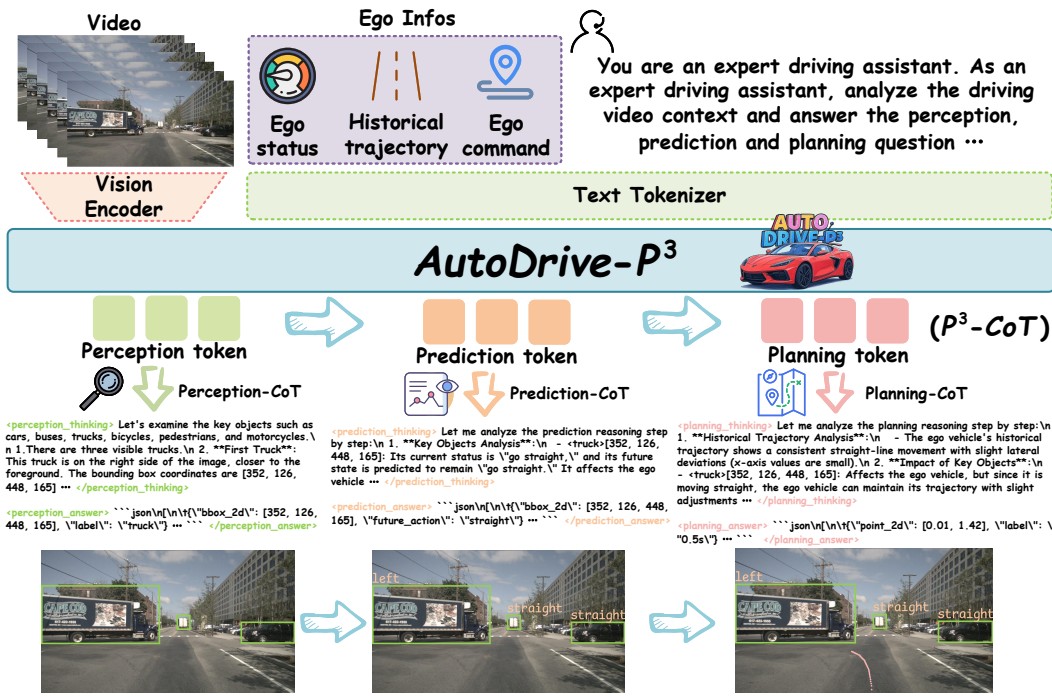

Figure 2: **Overview of** $AutoDrive\text{-}P^3$. It processes video and ego vehicle data through structured Perception-Prediction-Planning Chain-of-Thought ($P^3\text{-}CoT$) reasoning, generating interpretable step-by-step rationale and structured outputs for perception, prediction, and planning.

This formulation integrates all inputs to predict future positions sequentially, capturing temporal dependencies in a unified end-to-end framework.

**Group Relative Policy Optimization (GRPO).** GRPO improves learning stability by removing the dependence on a value function and optimizing a group-level, sample-wise objective (Shao et al., 2024). For each question-answer pair $(q, a)$, the behavior policy $\pi_{\theta_{\text{old}}}$ generates a group of $G$ responses $\{o_i\}_{i=1}^{G}$. The normalized advantage for the $i$-th response at step $t$ is computed as:

$$\hat{A}_{i,t} = \frac{R_i - \text{mean}(\{R_j\}_{j=1}^{G})}{\text{std}(\{R_j\}_{j=1}^{G})}, \tag{2}$$

where $R_i$ is the reward of the $i$-th response. The GRPO objective integrates a clipped surrogate loss with a KL penalty term:

$$\mathcal{J}_{\text{GRPO}}(\theta) = \mathbb{E}_{q, \{o_i\} \sim \pi_{\theta_{\text{old}}}(O|q)} \left[ \frac{1}{G} \sum_{i=1}^{G} \left( \mathcal{J}_i^R - \beta D_{\text{KL}}(\pi_\theta \| \pi_{\text{ref}}) \right) \right], \tag{3}$$

$$\mathcal{J}_i^R = \min\left( \frac{\pi_\theta(o_i|q)}{\pi_{\theta_{\text{old}}}(o_i|q)} A_i, \ \text{clip}\left( \frac{\pi_\theta(o_i|q)}{\pi_{\theta_{\text{old}}}(o_i|q)}, 1 - \epsilon, 1 + \epsilon \right) A_i \right). \tag{4}$$

By leveraging diverse responses sampled from the model itself, GRPO enhances the model's reasoning capability through exposure to varied reasoning paths and solutions.

## 4 METHODOLOGY

In this section, we propose the $AutoDrive\text{-}P^3$ framework, which integrates Perception, Prediction, and Planning for autonomous driving. Existing VLM-based datasets offer only fragmented QA pairs, unsuitable for GRPO training. To solve this, we create the $P^3\text{-}CoT$ dataset with unified CoT sequences linking the three tasks. We then perform supervised fine-tuning for cold-start initialization to align VLMs with the autonomous driving domain and generate accurate $P^3\text{-}CoT$ outputs. Finally, the $P^3\text{-}GRPO$ algorithm is introduced for post-training, providing hierarchical supervision and enabling collaborative optimization across modules to improve planning via iterative CoT reasoning.

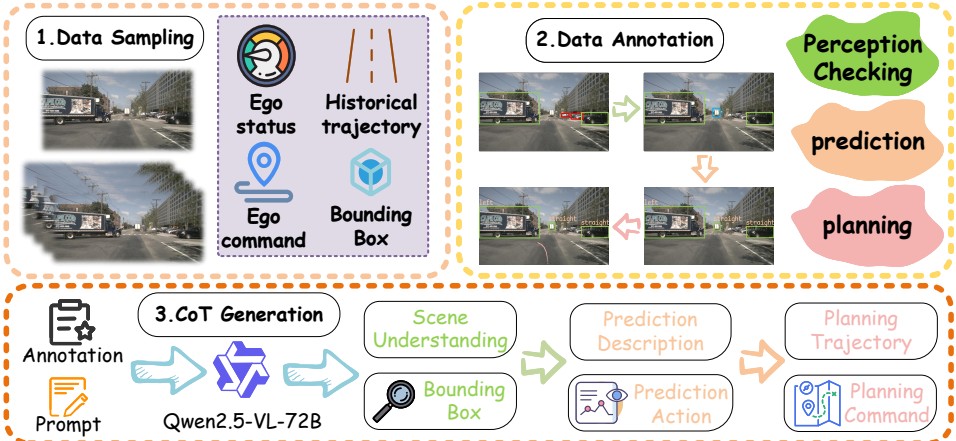

Figure 3: **The pipeline for constructing $P^3$-$CoT$ dataset.** We first sample data and annotations from existing datasets, then construct the labels of samples, focusing on key objects and using rule-based and manual filtering. Finally, with the help of advanced VLM, we construct the CoT, focusing on the connection among perception, prediction and planning three stages.

## 4.1   $P^3$-$CoT$ DATASET

To cover all the thinking steps of human drivers and meet the need of VLMs, a high quality reasoning dataset with key objects and detailed CoT annotations is strongly recommended. However, the following key challenges remain to be addressed: 1) lack of completed and comprehensive key object annotations, 2) requirements of unified chain of thought datasets for perception, prediction and plannning, 3) a proper CoT format suitable for VLM training instead of question-and-answear pairs. To address these issues, we propose $P^3$-$CoT$ dataset, a high-quality key objects' labels with CoT designed for VLM GRPO post-training, as shown in Fig. 3. We first identify and annotate critical objects in each key frame of the original dataset based on their potential impact on vehicle navigation, marking their bounding boxes as perception labels. We then derive prediction labels by projecting these critical objects' future trajectories. Finally, planning labels are obtained from the ego vehicle's planned trajectory. With these three-stage labels, we employ Qwen2.5-VL-72B (Bai et al., 2025) to generate coherent CoT data that seamlessly connects all three stages, with manual verification to ensure the correctness and logical integrity of the synthesized reasoning chains. Employing this annotation pipeline, we organize a high-quality and comprehensive CoT dataset with key object annotation and a unified $P^3$ arthchitecture in CoT format. $P^3$-$CoT$ includes 25303 frames from 850 scenes based on nuScenes, and 115434 frames from 1382 scenes based on NAVSIM. Additional description, statistics, and examples are attached in Appendix C.

Furthermore, we highlight that the proposed $P^3$-$CoT$ dataset benefits the model at both the holistic and modular levels. From a holistic perspective, the sequential reasoning process—from perception to prediction, and then to planning—guides the model in developing coherent and strategic driving behaviors. At the modular level, the specialized CoTs for perception, prediction, and planning respectively enhance the model's accuracy and reliability in executing each subtask. Comprehensive experiments in Section 5 validate these benefits across both levels.

## 4.2   SUPERVISED FINE-TUNING FOR COLD-START

To equip a VLM with autonomous-driving knowledge and structured reasoning capabilities, we conduct supervised fine-tuning (SFT) using the proposed $P^3$-$CoT$ dataset. As illustrated in Fig. 2, the model processes multimodal inputs $x = [x_{\text{ego}}; x_{\text{video}}; x_{\text{cmd}}; x_{\text{prompt}}]$ and learns to generate structured outputs organized into perception, prediction, and planning modules. The target output follows a unified format for each module:

$$y = [y_{\text{perception}}; y_{\text{prediction}}; y_{\text{planning}}], \quad \text{where} \quad y_{\text{module}} = [y_{\text{thinking}}; y_{\text{answer}}]. \tag{5}$$

This approach enables the model to produce coherent reasoning traces followed by concrete answers, establishing a foundational capability for Chain-of-Thought reasoning across all three autonomous

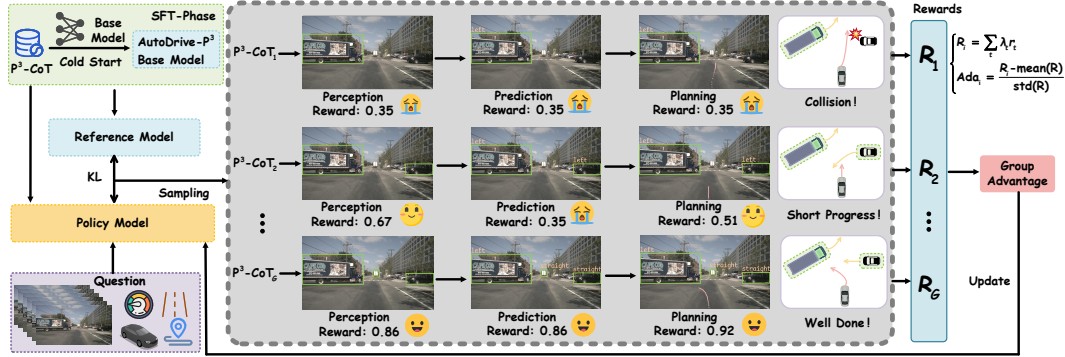

Figure 4: **The pipeline of** $P^3$**-**$GRPO$**.** We first cold start the base model using $P^3$-$CoT$ to make up for the gap between VLM and autonomous driving and learn the CoT answer format. Next we use GRPO to find the best optimization path and update our model.

driving stages. The training objective minimizes the negative log-likelihood of the target sequence:

$$\mathcal{L}_{\text{SFT}} = -\sum_{t=1}^{T} \log P(y_t \mid y_{<t}, x),$$ (6)

where $T$ is the total length of the target sequence. After cold-start SFT, the VLM acquires essential driving capabilities and produces interpretable $P^3$-$CoT$ outputs that enhance both transparency and performance, forming a solid basis for subsequent reinforcement learning.

### 4.3 $P^3$-$GRPO$ ALGORITHM

Following the cold-start SFT phase, we further enhance the VLM's reasoning capability across all three stages by applying the GRPO algorithm to the perception, prediction, and planning modules collectively, yielding the $P^3$-$GRPO$ algorithm, as shown in Fig. 4. Our approach employs a multi-component reward function to guide the policy model toward generating accurate, coherent, and well-structured outputs through coordinated reinforcement learning across these cognitive layers. The overall reward is computed as a weighted sum of the following components:

$$R(q, a) = \lambda_{\text{format}} \cdot R_{\text{format}} + \lambda_{\text{perc}} \cdot R_{\text{perc}} + \lambda_{\text{pred}} \cdot R_{\text{pred}} + \lambda_{\text{plan}} \cdot R_{\text{plan}},$$ (7)

where $\lambda_{\text{format}}$, $\lambda_{\text{perc}}$, $\lambda_{\text{pred}}$, and $\lambda_{\text{plan}}$ are weighting coefficients for each reward term. This integrated reward structure explicitly encodes the causal relationship between modules: perception enables prediction, and together they provide the necessary foundation for effective planning. By simultaneously optimizing together, our approach ensures that improvements in planning accuracy are grounded in corresponding enhancements in perceptual understanding and predictive capability.

**Perception Reward ($R_{\text{perc}}$)** measures object detection quality based on average IoU, precision ($P$), and recall ($R$), which encourages accurate and spatially precise perception, enabling reliable prediction and planning reasoning:

$$R_{\text{perc}} = \begin{cases} 1.0, & \text{if } |\mathcal{B}_{\text{gt}}| = 0 \text{ and } |\mathcal{B}_{\text{pred}}| = 0, \\ \text{IoU}_{\text{avg}} \cdot (0.5P + 0.5R), & \text{if } |\mathcal{B}_{\text{gt}}| > 0 \text{ and } |\mathcal{B}_{\text{pred}}| > 0, \\ 0.0, & \text{otherwise.} \end{cases}$$ (8)

**Prediction Reward ($R_{\text{pred}}$)** evaluates forecasting accuracy by combining behavior label correctness weighted by IoU and detection quality, which links perceptual accuracy with semantic correctness to foster robust prediction:

$$R_{\text{pred}} = \left( \frac{\sum_{(i,j)\in\mathcal{M}} \text{IoU}_{ij} \cdot \mathbb{I}(s_i = s_j)}{\sum_{(i,j)\in\mathcal{M}} \text{IoU}_{ij}} \right) \times \left( \text{IoU}_{\text{avg}} \cdot (0.5P + 0.5R) \right).$$ (9)

**Planning Reward ($R_{\text{plan}}$)** quantifies trajectory quality via L2 distance:

$$R_{\text{plan}} = \frac{2}{1 + e^{\text{clip}(L2, 0, L2_{\max})}}.$$ (10)

---

**Algorithm 1** $P^3$-$GRPO$: Perception-Prediction-Planning Group Relative Policy Optimization

---

**Require:** Policy model $\pi_\theta$ with $P^3$-$CoT$; dataset $\mathcal{D} = \{Q_n\}_{n=1}^N$; reward weights $\lambda_{\text{format}}$, $\lambda_{\text{perc}}$, $\lambda_{\text{pred}}$, $\lambda_{\text{plan}}$; KL constraint $\beta$; clip param $\epsilon$.
**Ensure:** Optimized policy model $\pi_\theta$

1: **for** iter = 1 to $N_{\text{RL}}$ **do**
2:     Sample query $Q \sim \mathcal{D}$
3:     Generate group of responses $\{a_i\}_{i=1}^M \sim \pi_\theta(\cdot|Q)$
4:     **for** $i = 1$ to $M$ **do**
5:        Parse $a_i$ into perception, prediction, planning components
6:        Compute rewards: $R_{\text{format}}^i$, $R_{\text{perc}}^i$, $R_{\text{pred}}^i$, $R_{\text{plan}}^i$
7:        Aggregate reward: $R_i = \lambda_{\text{format}} R_{\text{format}}^i + \lambda_{\text{perc}} R_{\text{perc}}^i + \lambda_{\text{pred}} R_{\text{pred}}^i + \lambda_{\text{plan}} R_{\text{plan}}^i$
8:     **end for**
9:     Compute mean $\bar{R} = \frac{1}{M}\sum_i R_i$ and std $\sigma_R = \sqrt{\frac{1}{M}\sum_i (R_i - \bar{R})^2}$
10:    **for** $i = 1$ to $M$ **do**
11:       Normalize advantage: $A_i = \frac{R_i - \bar{R}}{\sigma_R}$
12:    **end for**
13:    **for** $i = 1$ to $M$ **do**
14:       Compute ratio $r_i = \frac{\pi_\theta(a_i|Q)}{\pi_{\text{old}}(a_i|Q)}$
15:       Surrogate objective: $J_i = \min\left(r_i A_i, \text{clip}(r_i, 1-\epsilon, 1+\epsilon) A_i\right)$
16:    **end for**
17:    Compute policy loss: $\mathcal{L}_{\text{policy}} = -\frac{1}{M}\sum_i J_i$, KL penalty: $\mathcal{L}_{\text{KL}} = \beta D_{\text{KL}}(\pi_\theta || \pi_{\text{ref}})$
18:    Update $\pi_\theta$ via gradient descent on $\mathcal{L}_{\text{policy}} + \mathcal{L}_{\text{KL}}$
19: **end for**
20: **return** $\pi_\theta$

---

Table 1: **Performance comparison on nuScenes Benchmark.**

| Method | L2 (m) ↓ | | | | Collision (%) ↓ | | | | VLM |
|---|---|---|---|---|---|---|---|---|---|
| | 1s | 2s | 3s | Avg. | 1s | 2s | 3s | Avg. | |
| *Non-Autoregressive Methods* | | | | | | | | | |
| ST-P3 (Hu et al., 2022) | 1.33 | 2.11 | 2.90 | 2.11 | 0.23 | 0.62 | 1.27 | 0.71 | - |
| VAD (Jiang et al., 2023) | 0.17 | 0.34 | 0.60 | 0.37 | 0.07 | 0.10 | 0.24 | 0.14 | - |
| Ego-MLP (Li et al., 2024d) | 0.46 | 0.76 | 1.12 | 0.78 | 0.21 | 0.35 | 0.58 | 0.38 | - |
| UniAD (Hu et al., 2023) | 0.44 | 0.67 | 0.96 | 0.69 | 0.04 | 0.08 | 0.23 | 0.12 | - |
| InsightDrive (Song et al., 2025) | 0.23 | 0.41 | 0.68 | 0.44 | 0.09 | 0.10 | 0.27 | 0.15 | - |
| *Autoregressive Methods* | | | | | | | | | |
| GPT-Driver (Mao et al., 2023) | 0.20 | 0.40 | 0.70 | 0.44 | 0.04 | 0.12 | 0.36 | 0.17 | GPT-3.5 |
| DriveVLM (Tian et al., 2024) | 0.18 | 0.34 | 0.68 | 0.40 | 0.10 | 0.22 | 0.45 | 0.27 | Qwen2-VL-7B |
| OpenEMMA (Xing et al., 2025) | 1.45 | 3.21 | 3.76 | 2.81 | - | - | - | - | Qwen2-VL-7B |
| RDA-Driver (Huang et al., 2024a) | 0.17 | 0.37 | 0.69 | 0.40 | 0.01 | 0.05 | 0.26 | 0.10 | LLaVa-7B |
| OmniDrive (Wang et al., 2024) | **0.14** | **0.29** | 0.55 | **0.33** | 0.01 | **0.04** | 0.27 | 0.11 | LLava-7B |
| OpenDriveVLA (Zhou et al., 2025a) | **0.14** | 0.30 | 0.55 | **0.33** | 0.02 | 0.07 | 0.22 | 0.10 | Qwen2.5-VL-3B |
| AutoVLA (Zhou et al., 2025b) | 0.25 | 0.46 | 0.73 | 0.48 | 0.07 | 0.07 | 0.26 | 0.13 | Qwen2.5-VL-3B |
| AutoDrive-R² (Yuan et al., 2025) | 0.35 | 0.49 | 0.62 | 0.49 | - | - | - | - | Qwen2.5-VL-3B |
| *AutoDrive*-$P^3$ (Ours-Detailed) | 0.15 | 0.30 | **0.54** | **0.33** | **0.00** | **0.02** | **0.15** | **0.06** | Qwen2.5-VL-3B |
| *AutoDrive*-$P^3$ (Ours-Fast) | 0.16 | 0.31 | 0.56 | 0.34 | **0.00** | 0.04 | 0.20 | 0.08 | Qwen2.5-VL-3B |

These rewards together form a coordinated learning signal that promotes synergy among perception, prediction, and planning, ultimately driving accurate and interpretable autonomous driving behavior. The complete algorithmic procedure is summarized in Algorithm 1. Detailed formulations and comprehensive analyses of each reward component are provided in the Appendix D.

## 5 EXPERIMENTS

### 5.1 BENCHMARKS

**nuScenes** (Caesar et al., 2020). The nuScenes dataset comprises 1,000 real-world driving sequences. Following established evaluation protocols in related works (Hu et al., 2023; Jiang et al., 2023; Wang et al., 2024; Zhou et al., 2025a), we adopt two key metrics for planning performance: L2 displacement error and collision rate, using the same ST-P3 (Hu et al., 2022) metric settings.

**NAVSIM** (Dauner et al., 2024; Cao et al., 2025). To address the limited complexity of nuScenes, we further validate our approach using the NAVSIM benchmark. NAVSIMv1 (Dauner et al., 2024)

Table 2: **Performance comparison on NAVSIMv1 benchmark.**

| Method | Image | Lidar | NC↑ | DAC↑ | EP↑ | TTC↑ | Comf↑ | PDMS↑ |
|---|---|---|---|---|---|---|---|---|
| Human | ✗ | ✗ | 100.0 | 100.0 | 87.5 | 100.0 | 99.9 | 94.8 |
| Constant Velocity | ✗ | ✗ | 69.9 | 58.8 | 49.3 | 49.3 | 100.0 | 21.6 |
| Ego Status MLP | ✗ | ✗ | 93.0 | 77.3 | 62.8 | 83.6 | 100.0 | 65.6 |
| VADv2 (Weng et al., 2024) | ✔ | ✗ | 97.9 | 91.7 | 77.6 | 92.9 | 100.0 | 83.0 |
| UniAD (Hu et al., 2023) | ✔ | ✗ | 97.8 | 91.9 | 78.8 | 92.9 | 100.0 | 83.4 |
| LTF (Prakash et al., 2021) | ✔ | ✗ | 97.4 | 92.8 | 79.0 | 92.4 | 100.0 | 83.8 |
| TransFuser (Prakash et al., 2021) | ✔ | ✔ | 97.7 | 92.8 | 79.2 | 92.8 | 100.0 | 84.0 |
| PARA-Drive (Weng et al., 2024) | ✔ | ✗ | 97.9 | 92.4 | 79.3 | 93.0 | 99.8 | 84.0 |
| LAW (Li et al., 2024a) | ✔ | ✔ | 96.4 | 95.4 | 81.7 | 88.7 | 99.9 | 84.6 |
| DRAMA (Yuan et al., 2024) | ✔ | ✔ | 98.0 | 93.1 | 80.1 | 94.8 | 100.0 | 85.5 |
| Hydra-MDP (Li et al., 2024b) | ✔ | ✔ | 98.3 | 96.0 | 78.7 | 94.6 | 100.0 | 86.5 |
| DiffusionDrive (Liao et al., 2025) | ✔ | ✔ | 98.2 | 96.2 | 82.2 | 94.7 | 100.0 | 88.1 |
| WoTE (Li et al., 2025) | ✔ | ✔ | 98.5 | 96.8 | 81.9 | 94.9 | 99.9 | 88.3 |
| $AutoDrive\text{-}P^3$ (Ours-Detailed) | ✔ | ✗ | **99.1** | 97.4 | **84.8** | 96.5 | **100.0** | **90.6** |
| $AutoDrive\text{-}P^3$ (Ours-Fast) | ✔ | ✗ | 98.9 | **97.7** | 83.7 | **96.6** | 99.9 | 90.2 |

employs a simulation environment and uses the Predictive Driver Model Score (PDMS) for closed-loop evaluation. The PDMS is a composite metric defined as:

$$\text{PDMS} = \text{NC} \times \text{DAC} \times \left( \frac{5 \times \text{EP} + 5 \times \text{TTC} + 2 \times \text{Comf}}{12} \right), \tag{11}$$

where the components include No Collision (NC), Drivable Area Compliance (DAC), Ego Progress (EP), Time-to-Collision (TTC), and Comfort (Comf). In addition, NAVSIMv2 (Cao et al., 2025) offers a more comprehensive metric, named Extended Predictive Driver Model Score (EPDMS):

$$\text{EPDMS} = \text{NC} \times \text{DAC} \times \text{DDC} \times \text{TLC} \times \left( \frac{5 \times \text{EP} + 5 \times \text{TTC} + 2 \times \text{LK} + 2 \times \text{HC} + 2 \times \text{EC}}{16} \right), \tag{12}$$

where the components include Driving Direction Compliance (DDC), Traffic Light Compliance (TLC), Lane Keeping (LK), History Comfort (HC), Extended Comfort (EC) and other metrics are the same as PDMS. To reduce false positive penalties, NAVSIMv2 sets "human_penalty_filter" to true, disabling penalties when the human agent makes a violation; otherwise, it is set to false.

Table 3: **Performance comparison on NAVSIMv2 benchmark.**

| Method | NC↑ | DAC↑ | DDC↑ | TLC↑ | EP↑ | TTC↑ | LK↑ | HC↑ | EC↑ | EPDMS↑ False / True |
|---|---|---|---|---|---|---|---|---|---|---|
| Human | 100.0 | 100.0 | 99.8 | 100.0 | 87.4 | 100.0 | 100.0 | 98.1 | 90.1 | 90.3 / 94.5 |
| Ego Status MLP | 93.1 | 77.9 | 92.7 | 99.6 | 86.0 | 91.5 | 89.4 | 98.3 | 85.4 | 64.0 / - |
| Transfuser (Prakash et al., 2021) | 96.9 | 89.9 | 97.8 | 99.7 | 87.1 | 95.4 | 92.7 | 98.3 | 87.2 | 76.7 / 84.0 |
| HydraMDP++ (Li et al., 2024b) | 97.2 | 97.5 | 99.4 | 99.6 | 83.1 | 96.5 | 94.4 | 98.2 | 70.9 | 81.4 / - |
| DiffusionDrive (Liao et al., 2025) | 98.2 | 96.2 | **99.5** | 99.8 | 87.4 | 97.3 | **96.9** | 98.4 | **87.7** | 84.7 / 88.2 |
| WoTE (Li et al., 2025) | 98.5 | 96.8 | 98.8 | 99.8 | 86.1 | 97.9 | 95.5 | 98.3 | 82.9 | 84.2 / 87.7 |
| $AutoDrive\text{-}P^3$ (Ours-Detailed) | **99.1** | 97.4 | 99.2 | 99.8 | **88.0** | **98.7** | 96.3 | 98.3 | 85.5 | **86.2 / 89.9** |
| $AutoDrive\text{-}P^3$ (Ours-Fast) | 98.9 | **97.6** | 98.9 | 99.8 | 86.8 | 98.5 | 95.4 | 98.3 | 80.6 | 85.2 / 88.7 |

## 5.2 IMPLEMENTATION DETAILS

**nuScenes Benchmark.** Model inputs consist of 3-second video clips composed of 6 frames, drawn solely from the front-view camera. Images are resized to a resolution of 448×252. The ego-state information provided is only ego speed. The model is trained for 10 epochs with a batch size of 8.

**NAVSIM Benchmark.** Model input is constructed from 2-second video segments containing 4 frames, combining the front, front-left, and front-right camera views, which is then resized to 672×168. The ego-state information includes the longitudinal and lateral velocity and acceleration components (i.e., $v_x$, $v_y$, $a_x$, $a_y$). The model is trained for 10 epochs with a batch size of 32.

**Shared Settings.** We use Qwen2.5-VL-3B (Bai et al., 2025) as base model. All models are optimized using the AdamW optimizer across 8 A100 GPUs. During training with $P^3\text{-}GRPO$, 8

Table 4: **Ablation study on** $AutoDrive\text{-}P^3$ **on nuScenes Benchmark.**

| Method | Perception ↑ | Prediction ↑ | Planning (Collision, %) ↓ | | | |
|---|---|---|---|---|---|---|
| | | | 1s | 2s | 3s | Avg. |
| UniAD (Hu et al., 2023) | 0.32 | 0.31 | 0.04 | 0.08 | 0.23 | 0.12 |
| OmniDrive (Wang et al., 2024) | 0.37 | – | 0.01 | 0.04 | 0.27 | 0.11 |
| $AutoDrive\text{-}P^3$ (Only SFT) | 0.33 | 0.23 | 0.01 | 0.08 | 0.40 | 0.17 |
| $AutoDrive\text{-}P^3$ (SFT + Only Planning GRPO) | – | – | 0.03 | 0.08 | 0.24 | 0.12 |
| $AutoDrive\text{-}P^3$ (SFT + $P^3\text{-}GRPO$) | **0.64** | **0.54** | **0.00** | **0.02** | **0.15** | **0.06** |

Table 5: **Ablation study on different training setting on nuScenes benchmark.**

| Method | Group Size | History Traj. | Sensor Type | L2 (m) ↓ | | | | Collision (%) ↓ | | | |
|---|---|---|---|---|---|---|---|---|---|---|---|
| | | | | 1s | 2s | 3s | Avg. | 1s | 2s | 3s | Avg. |
| Ablation 1 | 4 | ✔ | Video | 0.17 | 0.32 | 0.65 | 0.38 | 0.01 | 0.06 | 0.30 | 0.13 |
| Ablation 2 | 8 | ✗ | Video | 0.17 | 0.33 | 0.68 | 0.39 | 0.02 | 0.07 | 0.33 | 0.14 |
| Ablation 3 | 8 | ✔ | Image | 0.16 | 0.32 | 0.61 | 0.36 | 0.01 | 0.05 | 0.26 | 0.12 |
| $P^3\text{-}GRPO$ | 8 | ✔ | Video | **0.15** | **0.30** | **0.54** | **0.33** | **0.00** | **0.02** | **0.15** | **0.06** |

$P^3\text{-}CoT$ samples are generated for each scenario. The reward function incorporates multiple components balanced by the following weights: $\lambda_{\text{format}}$, $\lambda_{\text{perc}}$, $\lambda_{\text{pred}}$, and $\lambda_{\text{plan}}$ in a ratio of 1:2:2:5. Following (Zhou et al., 2025b), we add PDMS to planning reward for NAVSIM benchmark. For each benchmark, we implement a dual-thinking setup consisting of a fast and a detailed version, as shown in Fig. 5. The fast version is designed for efficiency; while it adheres to the $P^3\text{-}CoT$ structure, it only yields the final answer from each module without reasoning. The detailed version, in contrast, provides the complete reasoning with answer for all modules.

## 5.3 COMPARISON WITH STATE-OF-THE-ART METHODS

As show in Table 1, we compare $AutoDrive\text{-}P^3$ with mainstream methods on nuScenes dataset. We achieve the same level as SOTA methods at L2 and overpass about 40% compared to SOTA methods at collision rare. In Table 2 and Table 3, $AutoDrive\text{-}P^3$ achieves the SOTA results with vision-only input on NAVSIMv1/v2 benchmark, achieving 90.6 PDMS and 89.9 EPDMS. Specifically, our method achieves comparable L2 scores with a significantly smaller model (Qwen2.5-3B vs. LLava-7B used in OmniDrive) and less training data (20k vs. 1000k samples used in OpenDriveVLA), while also attaining the best collision rate, demonstrating the superior efficiency and effectiveness of $AutoDrive\text{-}P^3$.

## 5.4 ABLATION STUDY

**Ablation Study on** $AutoDrive\text{-}P^3$**.** We conduct ablation studies on $AutoDrive\text{-}P^3$ with nuScenes to assess the effectiveness of joint RFT across perception–prediction–planning. We compare three settings—(1) Only SFT, (2) SFT + Only Planning GRPO, and (3) SFT + $P^3\text{-}GRPO$—against two end-to-end baselines: the small-scale UniAD and the VLM-based OmniDrive. As shown in Table 4, Only SFT already achieves a lower 2s collision rate than baselines. Adding Planning GRPO further reduces the 3s collision rate, matching baseline performance. Crucially, $P^3\text{-}GRPO$ yields large improvements in perception and prediction, surpassing all baselines and significantly boosting planning. These results demonstrate that $P^3\text{-}GRPO$ effectively captures the staged CoT across perception, prediction, and planning, leading to holistic gains in autonomous driving.

**Ablation Study on Training Settings.** We conduct additional ablation studies on three training configurations: GRPO group size, historical trajectory usage, and sensor modality. As shown in Table 5, results demonstrate that: (1) increasing group size from 4 to 8 enhances performance through more diverse reasoning samples; (2) incorporating historical trajectories improves contextual understanding; and (3) video sensors outperform image-based inputs by capturing temporal dynamics. Our full configuration achieves optimal results across all metrics.

**Runtime and Dual thinking modes.** We provide dual thinking modes' inference time compared to other methods, as shown in Fig. 5. We employ vLLM 0.8.0 (Kwon et al., 2023) acceleration on an H100 GPU, achieving near real-time performance (1 Hz).

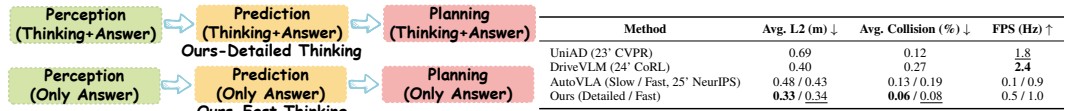

| Method | Avg. L2 (m) ↓ | Avg. Collision (%) ↓ | FPS (Hz) ↑ |
|---|---|---|---|
| UniAD (23' CVPR) | 0.69 | 0.12 | 1.8 |
| DriveVLM (24' CoRL) | 0.40 | 0.27 | 2.4 |
| AutoVLA (Slow / Fast, 25' NeurIPS) | 0.48 / 0.43 | 0.13 / 0.19 | 0.1 / 0.9 |
| Ours (Detailed / Fast) | **0.33** / 0.34 | **0.06** / 0.08 | 0.5 / 1.0 |

Figure 5: **Dual thinking modes and running time on nuScenes Benchmark.**

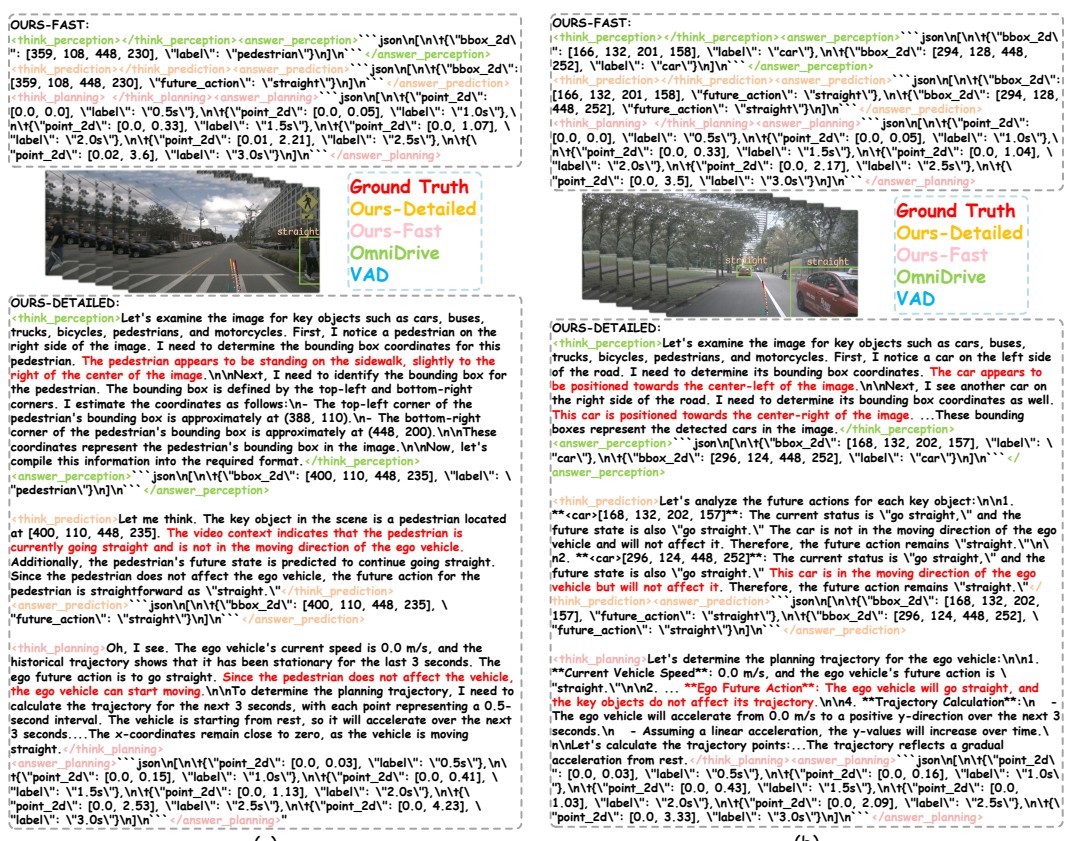

Figure 6: **Visualization.** Our model, taking into account the scenarios comprehensively, makes efficient plans that are both reasonable and safe.

## 5.5 VISUALIZATION

Fig. 6(a) demonstrates our method's ability to accurately perceive pedestrian location and predict safe passage opportunities, avoiding the overly conservative decisions of comparison methods. In Fig. 6(b), our approach successfully handles complex vehicle interactions by identifying key objects and their behaviors, producing trajectories that align with human driving habits.

## 6 CONCLUSION AND FUTURE WORK

In this work, we proposed $AutoDrive\text{-}P^3$, a novel VLM framework that establishes progressive connections between Perception, Prediction, and Planning. Our approach includes the $P^3\text{-}CoT$ dataset with unified reasoning chains and labels, supervised fine-tuning for domain adaptation, and the $P^3\text{-}GRPO$ algorithm for hierarchical multi-task supervision. Experiments on NAVSIMv1/v2 and nuScenes show state-of-the-art performance, reducing collision rate by 40% on nuScenes. To balance inference efficiency with performance, we introduce dual thinking modes: detailed thinking and fast thinking. Although $AutoDrive\text{-}P^3$ achieves state-of-the-art performance, it faces limitations in hallucinatory phenomena during reasoning. Additionally, our reinforcement learning is conducted in offline simulators, lacking interaction with real-world environments. Future work will focus on mitigating hallucinations, reducing inference time, and deploying the system in closed-loop settings.

## 7 ACKNOWLEDGMENTS

This work was supported by National Science and Technology Major Project (2024ZD01NL00101), Natural Science Foundation of China (62271013), Guangdong Provincial Key Laboratory of Ultra High Definition Immersive Media Technology (2024B1212010006), Guangdong Province Pearl River Talent Program (2021QN020708), Guangdong Basic and Applied Basic Research Foundation (2024A1515010155), Shenzhen Science and Technology Program (JCYJ20240813160202004, JCYJ20230807120808017, SYSPG20241211173440004), and was also financially supported by the Outstanding Talents Training Fund in Shenzhen. (*Corresponding author: Wei Gao*)

## 8 ETHICS STATEMENT

This work adheres to the ICLR Code of Ethics and all authors have read and adhered to the Code of Ethics. In this study, no human subjects is involved. The use of all datasets, including nuScenes (Caesar et al., 2020) and NAVSIM (Dauner et al., 2024; Cao et al., 2025), follows the relevant usage guidelines and public licenses, ensuring no violation of privacy. We have been careful to avoid any biased or discriminatory results during our research process. No personally identifiable information is used, and no privacy or security concerns will be raised due to our experiments. We are committed to maintaining transparency and integrity throughout the research process.

## 9 REPRODUCIBILITY STATEMENT

We have made every effort to ensure that the results presented in this paper are reproducible. All code and datasets have been made publicly available in an anonymous repository to facilitate replication and verification. The experimental setup, including training steps, model configurations, and hardware details, is described in detail in the paper. We have also provided a full description of $AutoDrive\text{-}P^3$ to assist others in reproducing our experiments.

Additionally, the datasets used in our experiments are publicly available, ensuring consistent and reproducible evaluation results.

We believe these measures will enable other researchers to reproduce our work and further advance the field.

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

## A    A USE OF LARGE LANGUAGE MODELS

We acknowledge the use of Large Language Models to assist in the preparation of this manuscript. We utilize Google's Gemini 2.5 Pro and DeepSeek-R1 for writing assistance. The specific applications are as follows:

- **Language and Readability:** To improve the grammar, clarity, and overall readability of the manuscript through language polishing.
- **Format Checking:** Gemini 2.5 Pro and DeepSeek-R1 are used to aid in improving grammatical fluency, and enhancing the overall readability of the text.

We emphasize that all scientific claims, hypotheses, experimental designs, results, analyses, and final conclusions are meticulously formulated, reviewed, and verified by the human authors. The authors take full and final responsibility for the entire content of this submission, in accordance with the ICLR policy.

## B    CONTRIBUTIONS

The core authors of this paper are Yuqi Ye, Zijian Zhang and Wei Gao. Yuqi Ye and Zijian Zhang are co-first authors. Wei Gao is the corresponding author. Yuqi Ye is responsible for conceiving the overall framework, developing the training and inference code, conducting the experiments, and performing data processing. Zijian Zhang is responsible for metrics evaluation, data visualization, data labeling, and data checking. All the authors participate in the discussion and writing of the paper.

## C    $P^3\text{-}CoT$ DATASET

In this section, we will provide a detailed introduction to $P^3\text{-}CoT$ dataset, including motivation, data collection, data composition and distribution and comparisons between datasets.

### C.1    MOTIVATION

The ultimate goal of autonomous driving models is to drive like humans. This target places high demands on models, requiring them to understand the driving environment and think like humans. Human drivers habitually identify and locate key objects that have a significant impact on driving decisions and actions during the driving, and subconsciously predict their future behaviors to help make the final driving routes. With deeper insight into autonomous driving models, the transformation of model architecture has undergone a shift from phased perception, prediction and planning to a unified end-to-end structure. Though end-to-end structures gradually become mainstream due to the reduction of accumulative errors, segmented consideration of perception, prediction and planning remains a core design concept when designing new modules. Meanwhile, in the process of human understanding the driving environment, paying too much attention to unimportant objects will instead reduce the concentration of human drivers and lead to unsafe driving behaviors. In other words, focusing only on key objects is necessary for efficient and safe driving behaviors, which is the same for the models. Considering the high requirements of comprehensive scene understanding ability and complex logical decision-making capability in autonomous driving, the vision language model (VLM) and reinforcement post-training show great potential. To cover all the thinking steps of human drivers and meet the need of VLMs, a high quality reasoning dataset with key objects and detailed CoT annotations is strongly recommended. This dataset should only focus on key objects and ignore unnecessary objects identification and localization, and construct a unified chain of thought (CoT) including perception, prediction and planning.

However, the existing datasets can not meet such requirements. Some object grounding datasets (Caesar et al., 2020) have been proposed to meet the need for fine-tuning VLMs in autonomous driving perception tasks, but too many objects requiring grounding will introduce additional noises as mentioned before. Existing approaches (Ding et al., 2024) attempt to establish chain of thought datasets to supervised fine-tuning VLMs, and some of them recognize the special meaning of key

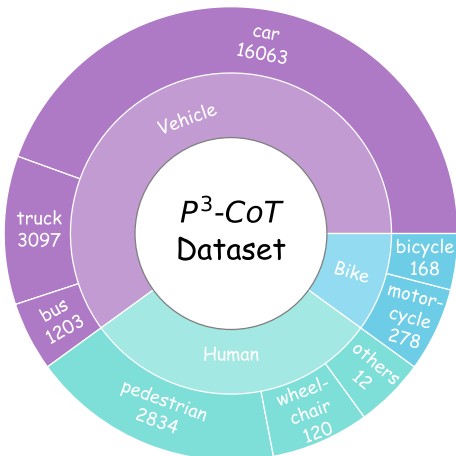

Figure 7: **The distribution of categories on** $P^3$-$CoT$ **(nuScenes).**

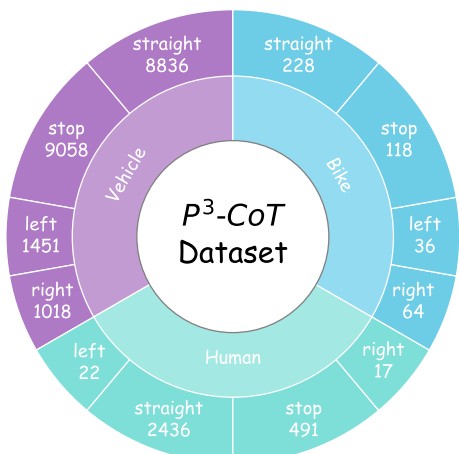

Figure 8: **The distribution of actions on** $P^3$-$CoT$ **(nuScenes).**

objects. These efforts only provide general descriptions of driving conditions, while overlooking the necessity of considering the driving process in stages and the connection between different stages. DriveLM (Sima et al., 2024) points out the importance of key objects and notices the significance of the connection among different thinking steps. But DriveLM lacks a clear definition of key objects and their impacts, and the annotations of DriveLM-nuScenes are incomplete both in key frame selection and key object localization. Moreover, though DriveLM uses graph to model the connection among the three stages of perception, prediction and planning, it formulates the data as question-answer pairs which do not match the labels used in the CoT format and remains fixed content templates lacking sufficient flexibility and diversity. Therefore, the following key challenges remain to be addressed: 1) lack of completed and comprehensive key object annotations, 2) requirements of unified chain of thought datasets for perception, prediction and plannning, 3) a proper CoT format suitable for VLM training instead of question-and-answear pairs.

## C.2 DATA COLLECTION

To address these issues, we propose $P^3$-$CoT$ dataset, a high-quality CoT dataset designed for VLM training and GRPO post-training. We first clarify significant concepts in our datasets. The key object of a specific scene is defined as those that human drivers will pay special attention in order to prevent potential dangerous events. This definition aims to simulate the attention of human driving, thereby aligning the model's focus with that of human drivers. The concepts of stages in our dataset are similar to end-to-end autonomous driving. The perception stage asks the model to localize targets in RGB images, but only key objects; the prediction stage utilizes the results from the perception stage to reason the future behaviors of key objects; the planning stage accepts the objects and corresponding actions from the first two stages and obtains the final waypoints after comprehensive consideration. To construct $P^3$ dataset, we first sample key frames of every scene in 2Hz following nuScenes settings, with 3 seconds history information as input and 3 seconds future trajectory as ground truth, and all the bounding boxes of objects will be projected into the image view. Considering that the distance between objects and ego car is an important factor in driving safety, we first filter out objects that are too far away and manually check the left objects to ensure high-quality key object annotations. It is worth noticing that we allow manual annotation of new objects that do not exist in previous dataset. After getting the key object annotations, we use trajectories of corresponding objects to determine future behaviors of objects in general directions like left or right and manually add the same labels according to videos if the object does not match any trajectory. And the way points in the future 3 seconds will be converted to the ego car coordinates as the ground truth. Given labels of the three stages and 3 seconds history information, we use the advanced Qwen2.5-VL-72B model to synthesize chain-of-thought data and make sure that only the results appearing in the previous stage can be used in the next stage to model the connections between different stages. The pipeline is shown in Fig. 3. Employing this annotation pipeline, we organize

Table 6: **The number of key objects in each frame of** $P^3$-$CoT$**.** The numbers on the table head are the number of key objects in each frame. Average is calculated as total number of key objects divided by total number of frames.

| Dataset | 0 | 1 | 2 | 3 | 4 | >4 | Average |
|---|---|---|---|---|---|---|---|
| $P^3$-$CoT$ (nuScenes) | 8187 | 9926 | 5121 | 1081 | 76 | 12 | 0.97 |
| $P^3$-$CoT$ (NAVSIM) | 24675 | 22915 | 20172 | 20794 | 20793 | 6085 | 2.08 |

Table 7: **The distribution of prediction actions of** $P^3$-$CoT$ **(NAVSIM).** The number of position of key objects in longitudinal and horizontal direction.

| $P^3$-$CoT$ (NAVSIM) | Left | Front | Right |
|---|---|---|---|
| **Far** | 7915 | 46909 | 4530 |
| **Middle** | 10601 | 33908 | 6558 |
| **Near** | 41435 | 31219 | 47815 |
| **Behind** | | 9137 | |

a high-quality and comprehensive CoT dataset with key object annotation and a unified perception-prediction-planning arthchitecture in CoT format. $P^3$-$CoT$ includes 19284 frames from 700 scenes in training set and 6019 frames from 150 scenes in test set based on nuScenes, and 103288 frames from 1192 scenes in training set and 12146 frames from 136 scenes in test set based on NAVSIM. With enough amount of CoT data, $P^3$-$CoT$ dataset can support VLM training for both supervised fine-tuning and reinforcement post-training, and benefit from the connections among stages, VLM trained by the dateset can gain advantages both from staged thought process and lower accumulated error of end-to-end autonomous driving, which also brings additional explainability to the black box in autonomous driving.

## C.3    DATA COMPOSITION

$P^3$-$CoT$ consists of two parts based on different benchmarks. On nuScenes, it includes a training set of 19,284 frames from 700 scenes and a test set of 6,019 frames from 150 scenes. On NAVSIM, it includes a training set of 103,288 frames from 1,192 scenes and a test set of 12,146 frames from 136 scenes. All frames are annotated with detailed information across perception, prediction, and planning stages, with explicit connections among the three tasks.

For perception stage, we have tallied up the number of key objects that occur in each frame in Table 6 and the distribution of future actions of objects in Fig. 8 and the distribution of categories in Fig. 7. Considering the relatively simple road conditions of the nuScenes dataset, the key objects gathered from each frame should be sparse. The results listed in Table 6 show that our pipeline for selecting key objects is reasonable and successfully reduces the additional and unnecessary objects in the scene. The distribution of categories presents the diversity of $P^3$-$CoT$ and that of actions is in line with the situation in reality.

For prediction stage and planning stage, we annotate the future action of every object, including the ego vehicle and other vehicles, and give the detailed results in Table 8, Table 7 and Fig. 8. For nuScenes, future actions of prediction stage and planning stage have a similar distribution, but the action straight of planning stage is more than that of prediction stage due to requirements of data collection. For NAVSIM, we keep the same command types as nuScenes and adapt prediction action types to NAVSIM. The actions of objects in NAVSIM focus on near range and front direction, showing the complexity of NAVSIM in our settings.

## C.4    COMPARISONS BETWEEN DATASETS

To highlight the advantages of $P^3$-$CoT$ dataset, we give comprehensive comparisons of our dataset and others in Table 9. We can tell from the table that $P^3$-$CoT$ dataset annotates a substantial number of frames and includes perception, prediction and planning all three stages. Unlike other

Table 8: **The distribution of planning commands in $P^3$-$CoT$.** The number of planning commands of ego vehicle.

| Planning Commands | Left | Right | Stop | Straight |
|---|---|---|---|---|
| $P^3$-$CoT$ (nuScenes) | 1299 | 1627 | 4919 | 16558 |
| $P^3$-$CoT$ (NAVSIM) | 27293 | 13524 | 1832 | 72785 |

Table 9: **The comparisons of existing datasets of scale and structure.** Source Dataset: Source of data sampled. Frames: the number of frames labeled by methods. Perception, Prediction, Planning: whether the dataset includes the information about perception, prediction and planning. Type: data organization format.

| Dataset | Source Dataset | Frames | Perception | Prediction | Planning | Type |
|---|---|---|---|---|---|---|
| nuScenes-QA (Qian et al., 2024) | nuScenes | 34149 | ✔ | ✗ | ✗ | QA |
| nuInstruct (Ding et al., 2024) | nuScenes | 11850 | ✔ | ✔ | ✔ | QA |
| nuPrompt (Wu et al., 2025) | nuScenes | 34149 | ✔ | ✗ | ✗ | QA |
| DriveLM-nuScenes (Sima et al., 2024) | nuScenes | 4871 | ✔ | ✔ | ✔ | QA |
| LingoQA (Marcu et al., 2024) | LingoQA | 28000 | — | — | — | QA |
| DRAMA (Malla et al., 2023) | DRAMA | 17785 | ✔ | ✗ | ✔ | QA |
| $P^3$-$CoT$(Ours) | nuScenes | 24403 | ✔ | ✔ | ✔ | CoT+Label |
| | NAVSIM | 114348 | ✔ | ✔ | ✔ | CoT+Label |

datasets, $P^3$-$CoT$ formulates the data in CoT format and maintains the close connection among stages, owning the special advantages to combine both staged interpretation and unified training process.

## C.5 DATA QUALITY ASSESSMENT

To ensure the high quality of the $P^3$-$CoT$ dataset, we adopt a manual assessment protocol through sampling inspection, following DriveLM (Sima et al., 2024), and other quality assessment (Sun et al., 2025a; 2024; 2025b). The evaluation is conducted at both holistic and modular levels. At the holistic level, each CoT label is manually inspected to ensure it strictly follows the prescribed reasoning structure—progressing completely and sequentially from perception to prediction and then to planning, with all final module outputs present and correctly formatted. At the modular level, we perform a fine-grained manual check for factual consistency and reasoning quality. The dataset is divided into 10 splits, each assigned to three independent annotators. They randomly sample 10% of their assigned split and meticulously examine the alignment between the generated CoT and the ground-truth labels for every module. This includes assessing the factual consistency of the reasoning steps, the fluency of the language, and the overall logical plausibility. In both levels of inspection, each reviewed CoT is labeled as "good" or "poor". The accuracy for a split is calculated based on the proportion of "good" samples. We set a passing accuracy threshold of 90% for a split to be considered qualified. Any split failing to meet this standard is deemed imperfect, and the CoT data for that portion is regenerated and re-submitted for assessment. This iterative process continues until all splits pass the manual quality check by all annotators.

## D REWARD SETTING

**Perception Reward** ($R_{\mathbf{perc}}$): This reward measures the quality of key object detection. Let $\mathcal{B}$pred and $\mathcal{B}$gt denote the sets of predicted and ground-truth bounding boxes, respectively. The reward is based on the average IoU of matched boxes, precision ($P$), and recall ($R$):

$$R_{\text{perc}} = \begin{cases} 1.0, & \text{if } |\mathcal{B}_{\text{gt}}| = 0 \text{ and } |\mathcal{B}_{\text{pred}}| = 0, \\ \text{IoU}_{\text{avg}} \cdot (0.5P + 0.5R), & \text{if } |\mathcal{B}_{\text{gt}}| > 0 \text{ and } |\mathcal{B}_{\text{pred}}| > 0, \\ 0.0, & \text{otherwise.} \end{cases} \tag{13}$$

The perception reward serves three crucial purposes: First, it ensures comprehensive scene understanding by penalizing both false positives (detecting non-existent objects) and false negatives (missing actual objects). Second, it maintains spatial accuracy through the IoU metric, guaranteeing that detected objects are precisely localized. Third, it provides immediate and interpretable feedback to the model about its perceptual capabilities, enabling focused improvement in visual understanding.

This structured reward mechanism forces the model to develop sophisticated visual reasoning skills, including object recognition, spatial relationships understanding, and scene context interpretation. By explicitly rewarding accurate perception, we create a solid foundation upon which prediction and planning modules can build reliable and safe driving strategies. The perception reward thus acts as a crucial enabler for the entire cognitive pipeline, ensuring that subsequent decisions are based on a veridical representation of the driving environment.

**Prediction Reward ($R_{\mathbf{pred}}$):** This component evaluates the model's ability to accurately forecast the future behavior of detected objects, serving as the critical bridge between perceptual understanding and planning decisions. The reward function is carefully designed to integrate both spatial localization accuracy and behavioral semantics:

$$R_{\text{pred}} = \left( \frac{\sum_{(i,j) \in \mathcal{M}} \text{IoU}_{ij} \cdot \mathbb{I}(s_i = s_j)}{\sum_{(i,j) \in \mathcal{M}} \text{IoU}_{ij}} \right) \cdot \left( \text{IoU}_{\text{avg}} \cdot (0.5P + 0.5R) \right), \tag{14}$$

where $\mathcal{M}$ represents the set of successfully matched prediction-ground truth pairs, and $\mathbb{I}$ is the indicator function that returns 1 when the predicted action label $s_i$ matches the ground truth $s_j$, and 0 otherwise.

The formula consists of two multiplicative components. The first term calculates a weighted behavior accuracy score, where each matched pair's label correctness is weighted by its IoU value. This design ensures that predictions with better spatial alignment contribute more significantly to the reward. The second term represents the fundamental detection quality, computed as the product of average IoU and the F1-score (harmonic mean of precision and recall), which maintains the basic requirement of accurate object detection and tracking.

This reward design serves two crucial purposes. First, it explicitly encodes the dependency between accurate perception and reliable prediction - the model cannot achieve high prediction rewards without first establishing solid perceptual foundations. Second, it emphasizes that behavioral prediction quality is intrinsically tied to spatial accuracy; even correct action labels receive reduced rewards if the associated bounding boxes are poorly localized.

By designing the prediction reward in this manner, we force the model to develop a comprehensive understanding of scene dynamics, where it must not only identify objects correctly but also anticipate their future behaviors accurately. This approach ensures that the prediction module provides meaningful and reliable inputs to the planning system, enabling the generation of safe and efficient driving strategies that account for the predicted evolution of the traffic environment.

**Planning Reward ($R_{\mathbf{plan}}$):** This component serves as the ultimate performance metric that evaluates the quality of the ego vehicle's planned trajectory, representing the final output of the entire reasoning pipeline. The reward is calculated through an exponential transformation of the L2 distance between the predicted trajectory points and their ground-truth counterparts:

$$R_{\text{plan}} = \frac{2}{1 + e^{\text{clip}(\text{L2}, 0, L2_{\text{max}})}}, \tag{15}$$

where L2 represents the mean Euclidean distance between corresponding points in the predicted and ground-truth trajectories across all future time horizons. Following AutoVLA (Zhou et al., 2025b), we add PDMS to planning reward for NAVSIM benchmark.

The planning reward serves as the ultimate validator of the entire $P^3$ reasoning chain. While high rewards in perception and prediction are necessary prerequisites, they are insufficient without corresponding excellence in planning. This design explicitly teaches the model that accurate perception and reliable prediction are valuable precisely because they enable superior planning decisions. The planning reward thus creates a powerful end-to-end learning signal that backpropagates through all

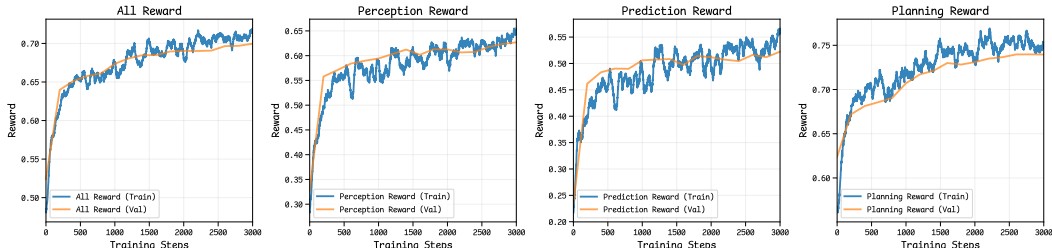

Figure 9: **Training and testing rewards for perception, prediction, planning, and all reward.** The results shows the consistent improvement in all rewards, which proves the tight inter-connection among three stages and the effectiveness of $P^3$-$CoT$.

modules, encouraging the development of coordinated representations where each component works synergistically toward the final goal of generating safe, comfortable, and efficient driving trajectories.

By placing the planning reward at the apex of our reward hierarchy, we ensure that the model optimizes not for intermediate metrics but for the ultimate objective of successful autonomous navigation, while maintaining the interpretability and safety guarantees provided by the structured $P^3$-$CoT$ reasoning process.

These rewards with $P^3$-$GRPO$ algorithm ensure that improvements in planning performance are grounded in corresponding enhancements in perceptual understanding and predictive capability, creating a synergistic effect where each module's optimization contributes to the overall driving performance. The algorithm maintains the interpretability and safety guarantees provided by the structured $P^3$-$CoT$ reasoning process while achieving superior autonomous driving performance through multi-module reinforcement learning.

**Reward Visualization.** We visualize the training and testing rewards for perception, prediction, planning, and the all reward, as shown in Fig. 9. The results demonstrate that through training with our $P^3$-$CoT$ dataset and $P^3$-$GRPO$ algorithm, the three modules exhibit mutual reinforcement, leading to consistent improvement in their respective rewards. This synergistic effect is particularly beneficial for planning performance, as the enhanced perception and prediction capabilities provide more reliable inputs for trajectory generation. The progressive optimization across modules ensures coherent reasoning and decision-making, ultimately contributing to more robust autonomous driving performance.

**Ablation Study on Reward Weight.** As shown in Table 10, we conduct additional ablation studies on three reward weight configurations. Results indicate that an unbalanced setting (e.g., 1:1:1:7), which overemphasizes the planning reward, can hinder the optimization of perception and prediction modules. This imbalance ultimately leads to inferior overall planning performance, as accurate planning is contingent upon reliable inputs from the preceding stages.

Table 10: **Weight Setting on nuScenes Benchmark.**

| Reward weight | Perception ↑ | Prediction ↑ | Planning (Avg. L2) ↓ |
|---|---|---|---|
| 1:2:2:5 | **0.64** | **0.54** | **0.33** |
| 1:1:1:7 | 0.63 | 0.53 | 0.34 |

# E    EXPERIMENTAL SETUP

We conduct experiments on two autonomous driving benchmarks: nuScenes and NAVSIM. The detailed experimental configurations are summarized in Table 11. For both datasets, we compare the Cold-Start baseline with our proposed $P^3$-$GRPO$ approach under consistent data settings. We also perform an ablation experiment by removing the KL divergence term from the training objective. As demonstrated in Fig. 10, the model without KL regularization suffers from significant performance degradation as training progresses, eventually leading to model collapse. This occurs because the absence of KL constraint allows the model to deviate excessively from the base policy, resulting in

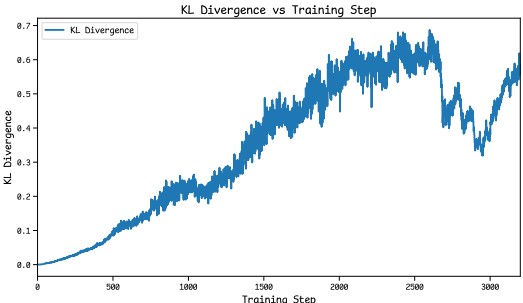

Figure 10: **Ablation Study on KL Divergence.**

unstable optimization. Therefore, we recommend retaining the KL divergence term during training to ensure the model maintains reasonable proximity to the base policy while improving performance.

Table 11: **Experimental setup.**

|  | nuScenes | | NAVSIM | |
|---|---|---|---|---|
|  | Cold-Start | $P^3$-$GRPO$ | Cold-Start | $P^3$-$GRPO$ |
| **Data Setting** | | | | |
| Video Shape | [6, 3, 252, 448] | [6, 3, 252, 448] | [4, 3, 168, 672] | [4, 3, 168, 672] |
| History Traj | 6 | 6 | 4 | 4 |
| Future Traj | 6 | 6 | 8 | 8 |
| Ego Infos | Only $V$ | Only $V$ | $a_x, a_y, v_x, v_y$ | $a_x, a_y, v_x, v_y$ |
| **Optimization** | | | | |
| Epoch | 1 | 5 | 2 | 10 |
| Batch size | 8 | 256 | 8 | 256 |
| Optimizer | AdamW | AdamW | AdamW | AdamW |
| Learing Rate | 2e-5 | 1e-6 | 2e-5 | 1e-6 |
| **GRPO Setting** | | | | |
| Group Size | – | 8 | – | 8 |
| KL Weight | – | 0.01 | – | 0.01 |

## F   QUALITATIVE COMPARISON OF TRAJECTORY PLANNING

To further explain the advantages of our method, we provide an intuitive comparison of the results in this section. For nuScenes samples, the red points denote ground truth trajectory, the orange denotes our trajectory with detail CoT, the pink denotes our trajectory with only CoT framework, the green denotes OmniDrive trajectory and the blue denotes VAD trajectory. For NAVSIM samples, the red points denote ground truth trajectory, the orange denotes our trajectory with detail CoT, the pink denotes our trajectory with only CoT framework, the green denotes WoTE trajectory and the blue denotes DiffusionDrive trajectory. Due to camera projection limitations, too short and too deviated trajectories will not appear in the images, such as stop situations. The CoT and answers corresponding to the specific sample are shown on the right of the figure.

In Fig. 11(a), the front view shows a scene at night with waiting cars. Due to image distortion, the street lamps have shifted towards the green light to a certain extent. The ground truth trajectory is to stop and wait, while the trajectories of comparison methods ignore the front car and move forward mistakenly. Our method first correctly observes the key objects, the closest two cars, and does not misunderstand the meaning of the lights, and then precisely gives the right future actions of the two. This result supports our planning decisions and our method finally takes the same actions as ground truth. This shows our method the powerful ability of scene understanding.

In Fig. 11(b), two trucks are parked by the roadside, and one of them blocks moving direction of the ego car. The ground truth trajectory still moves forward and tends to return to the initial road, but it seems too close to the truck on the right. The comparison methods do not work well in this sample. Our method can still identify key objects as before and provide good prediction answers. Though

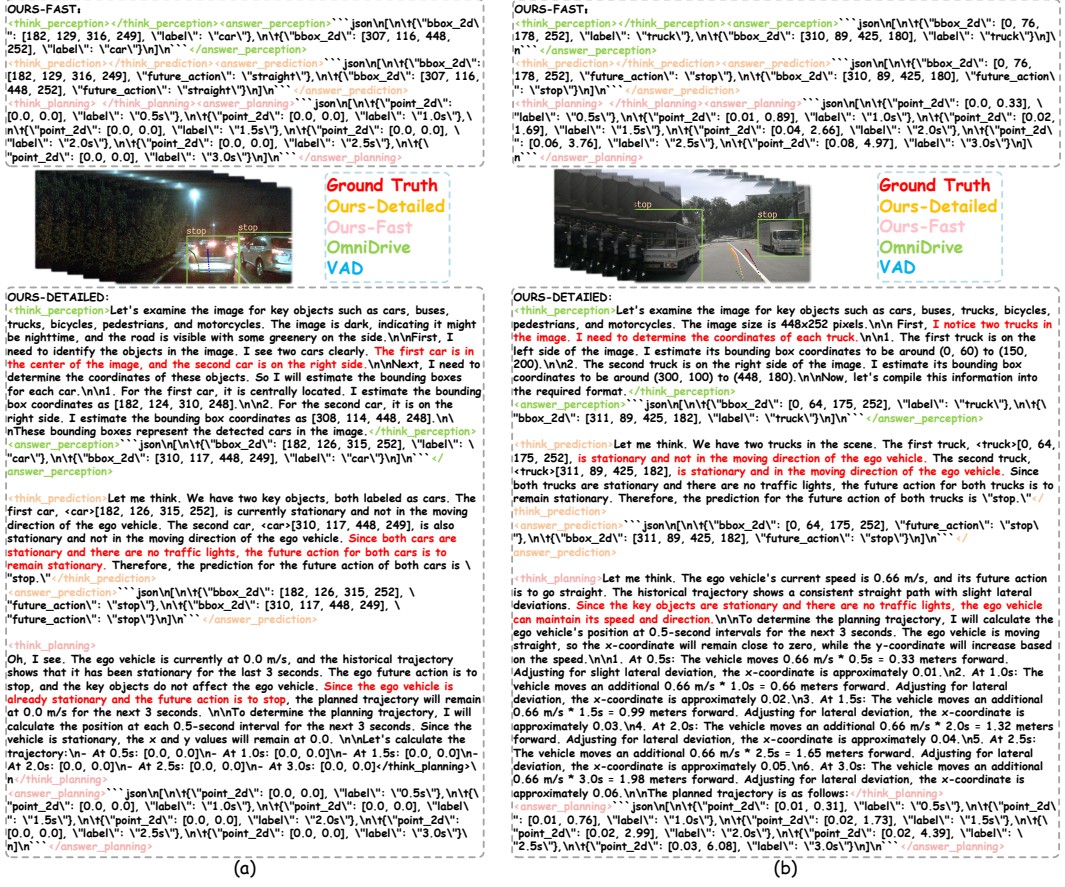

Figure 11: **Visualization examples of specific samples.** Every example is equipped with visualization of results on the top and CoT answer on the bottom.

the trajectory is at a certain distance from the ground truth, we believe that our method takes into account the truck on the right to correct its initial decision and avoid potential security issues.

Except for the specific explanation of the samples, we also provide special cases with completed questions and answers in Fig. 12 to Fig. 17. The first two are nuScenes samples and the last four are NAVSIM samples.

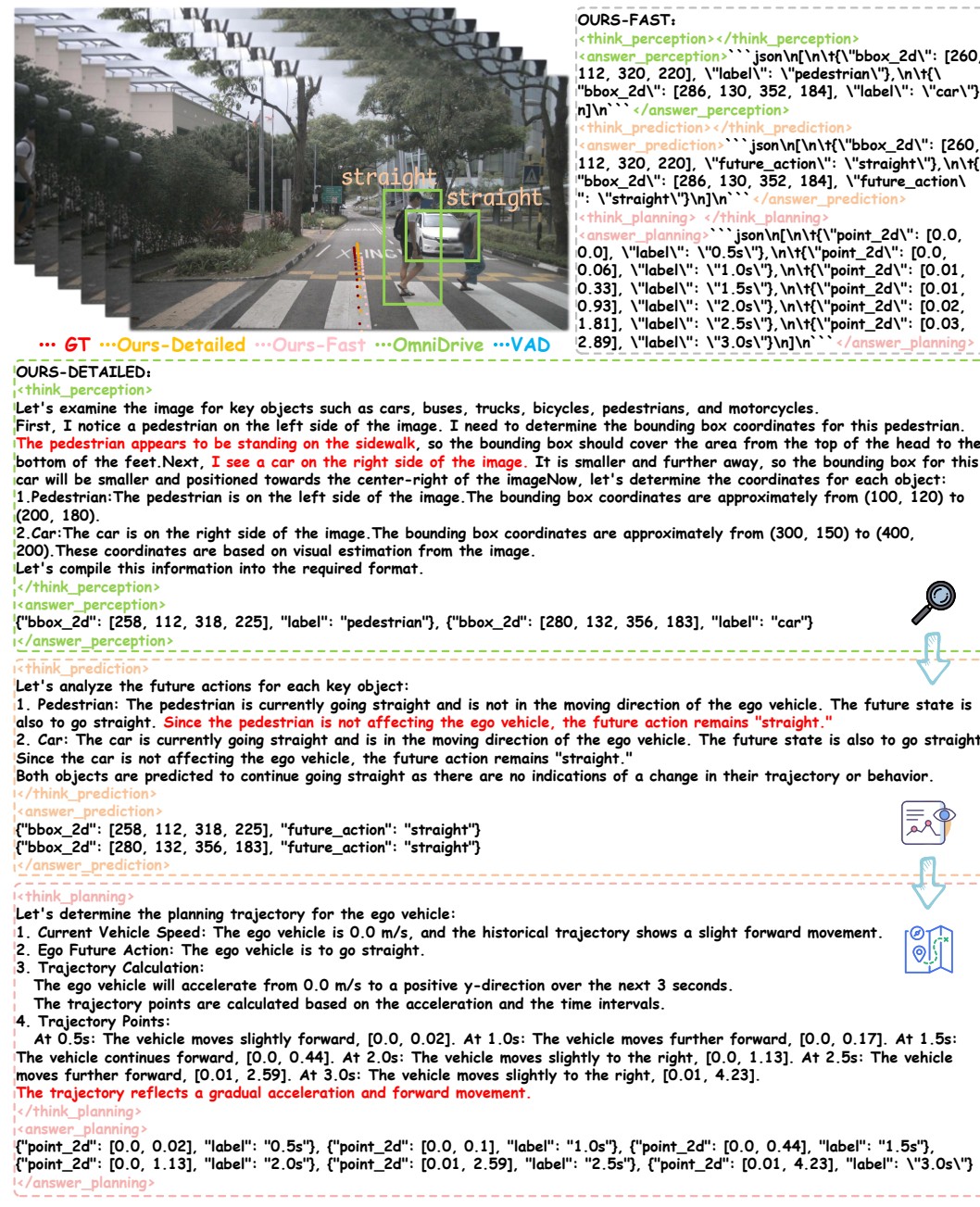

Figure 12: **Visualization nuScenes cases with completed questions and answers.** $AutoDrive\text{-}P^3$ successfully identifies and localizes the key objects, giving the correct actions. Based on these judgments, our method makes the efficiency planning decision in this sample.

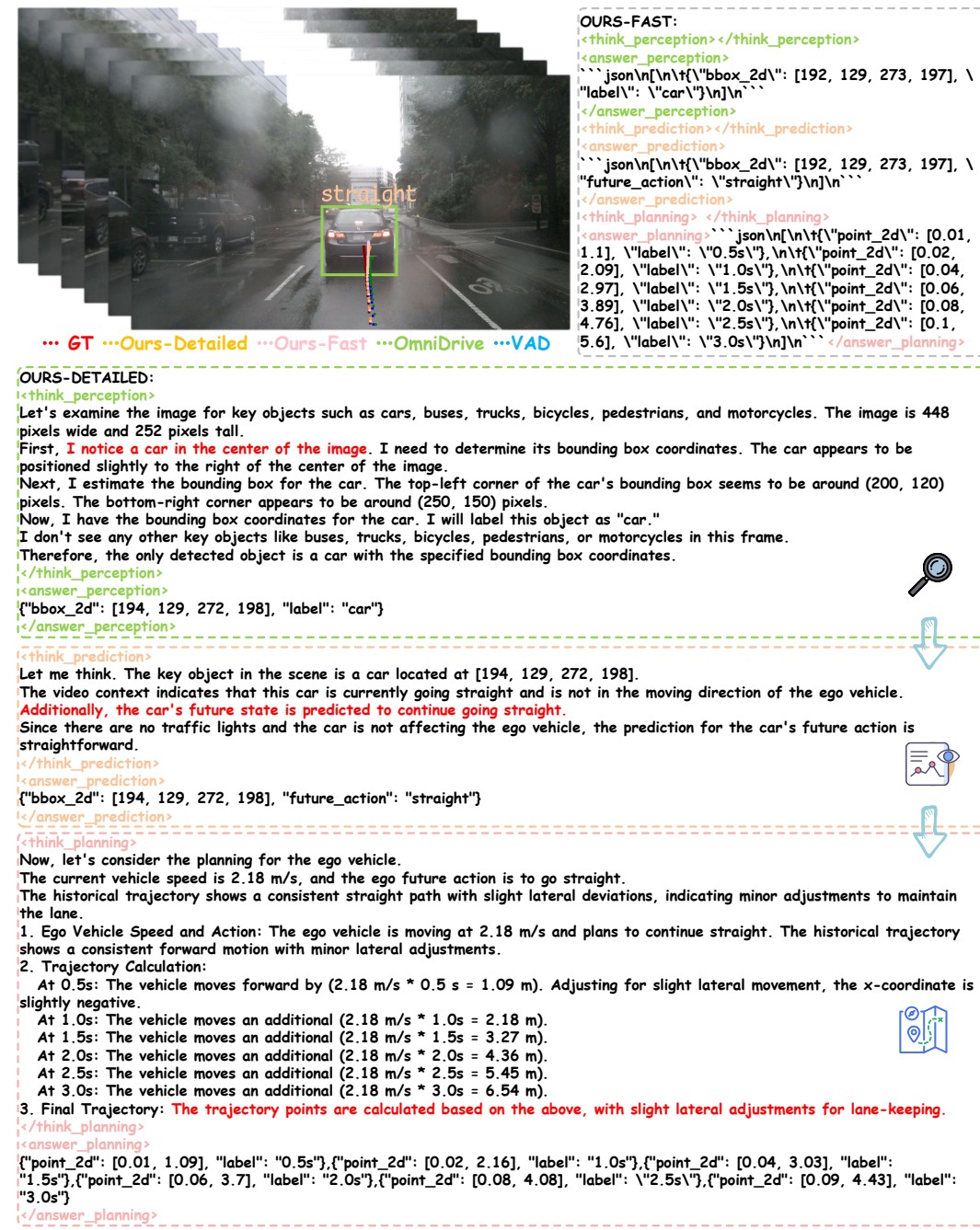

Figure 13: **Visualization nuScenes cases with completed questions and answers.** $AutoDrive\text{-}P^3$ successfully recognizes the driving command and provides the best trajectory instead of conservative one compared with other method.

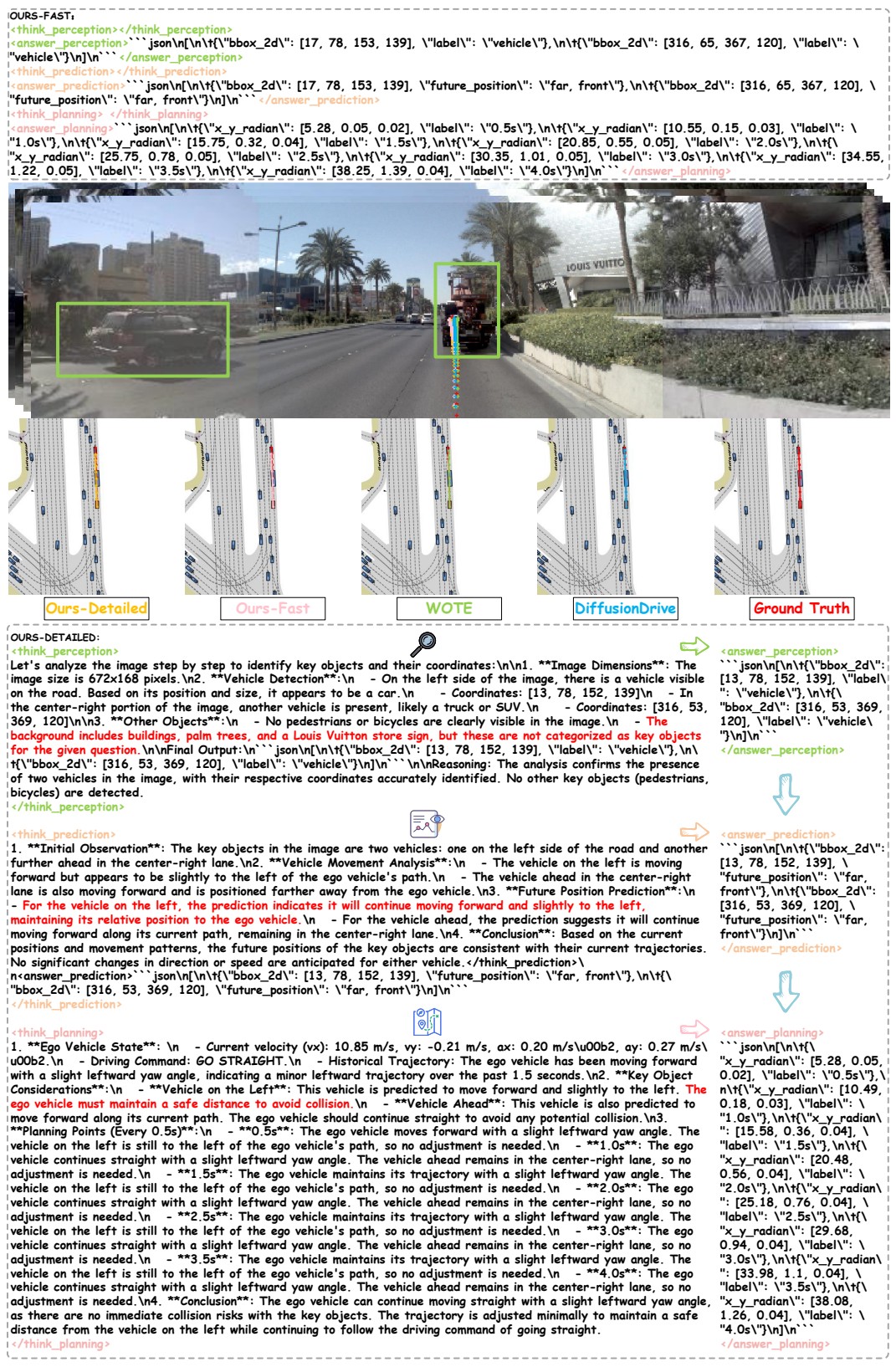

Figure 14: **Visualization NAVSIM cases with completed questions and answers.** *AutoDrive-P³* successfully predicts the future action of the truck and follows the forward vehicle in a safe distance.

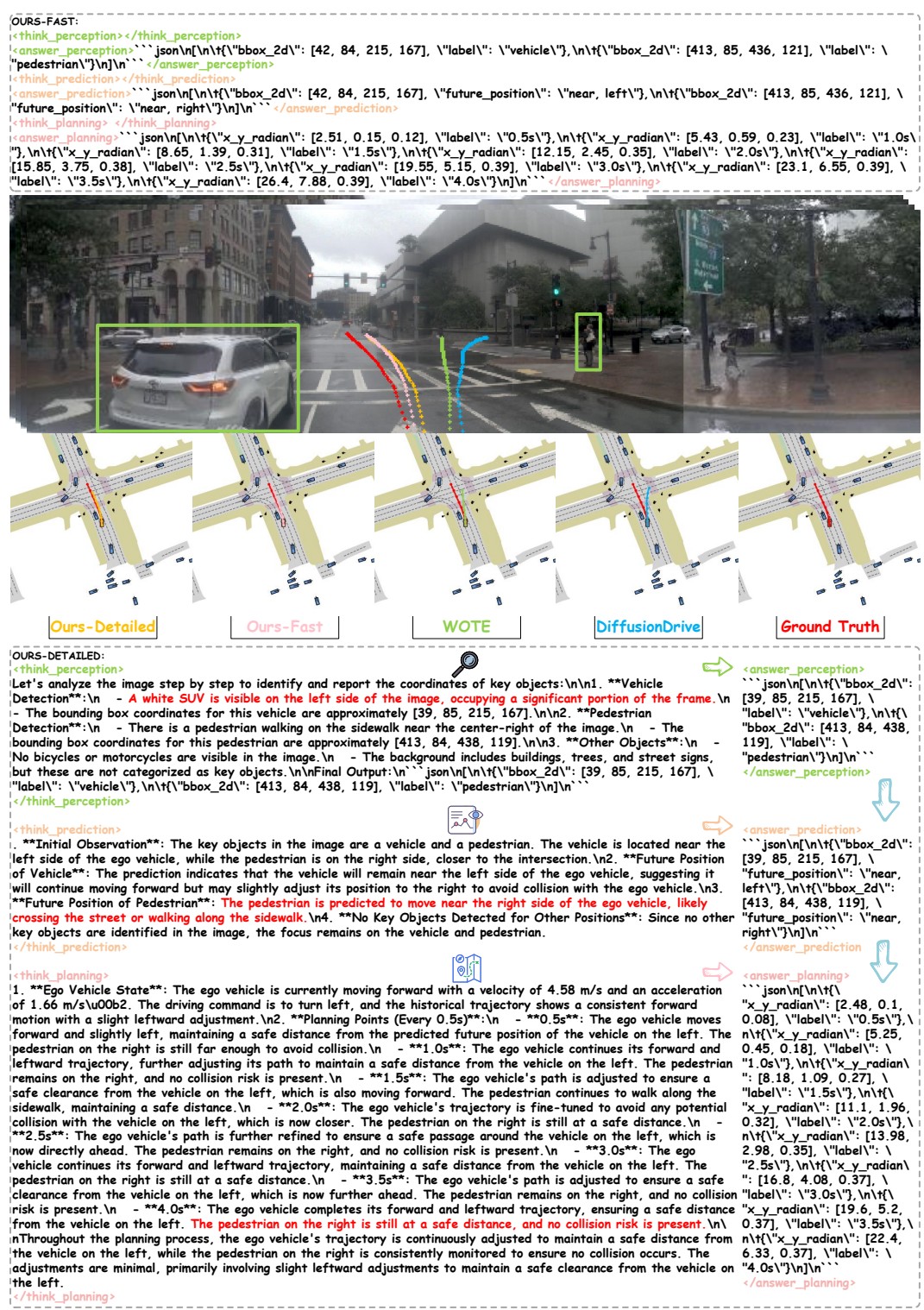

Figure 15: **Visualization NAVSIM cases with completed questions and answers.** *AutoDrive-P*$^3$ successfully locates the key object on the left and take a appropriate lane change action to move forward, while other methods provide a wrong trajectories to turn right.

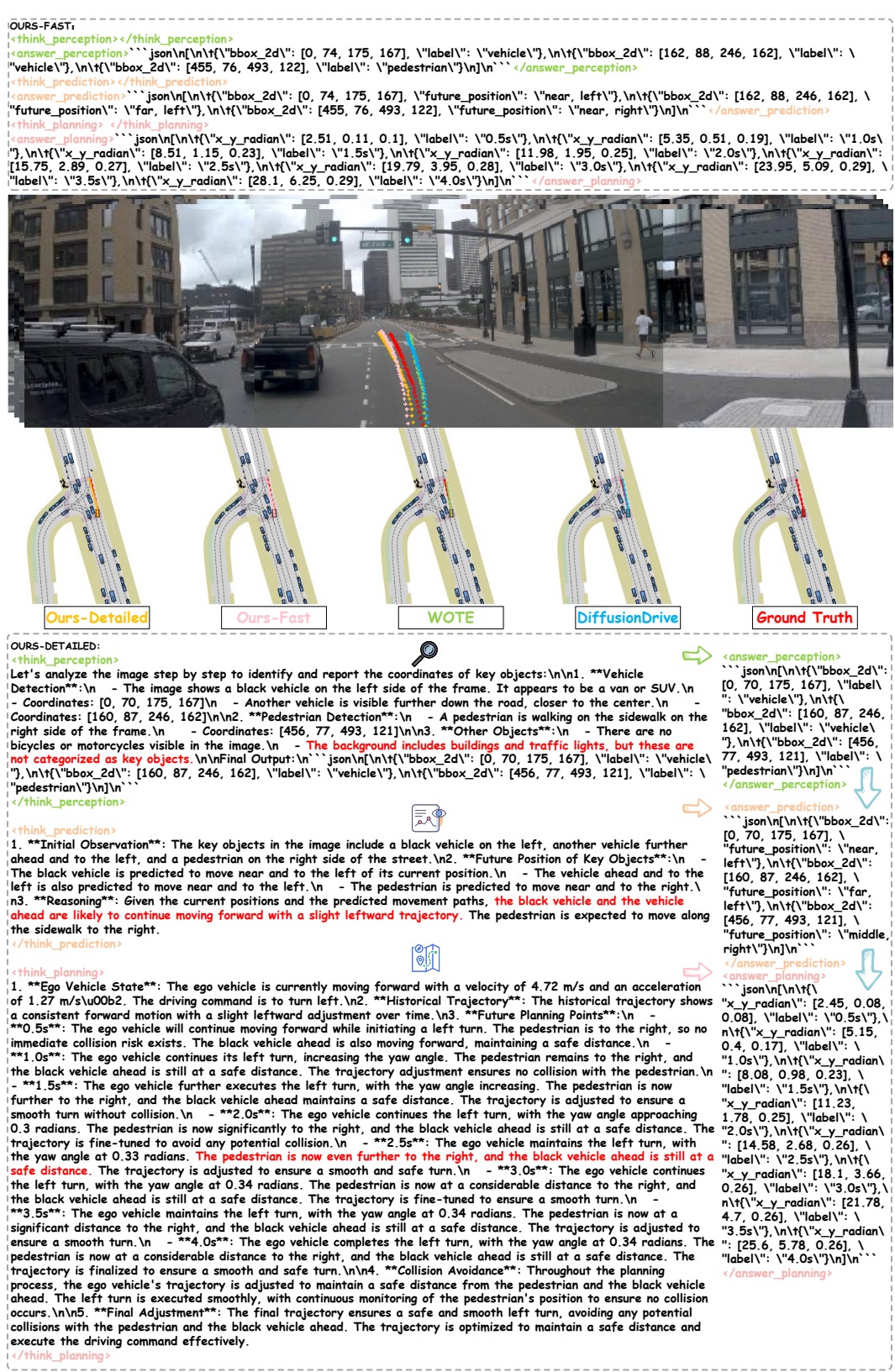

Figure 16: **Visualization NAVSIM cases with completed questions and answers.** $AutoDrive\text{-}P^3$ successfully locates key objects on the left and take a appropriate lane change action to move forward, while other methods drive into an illegal driving area.

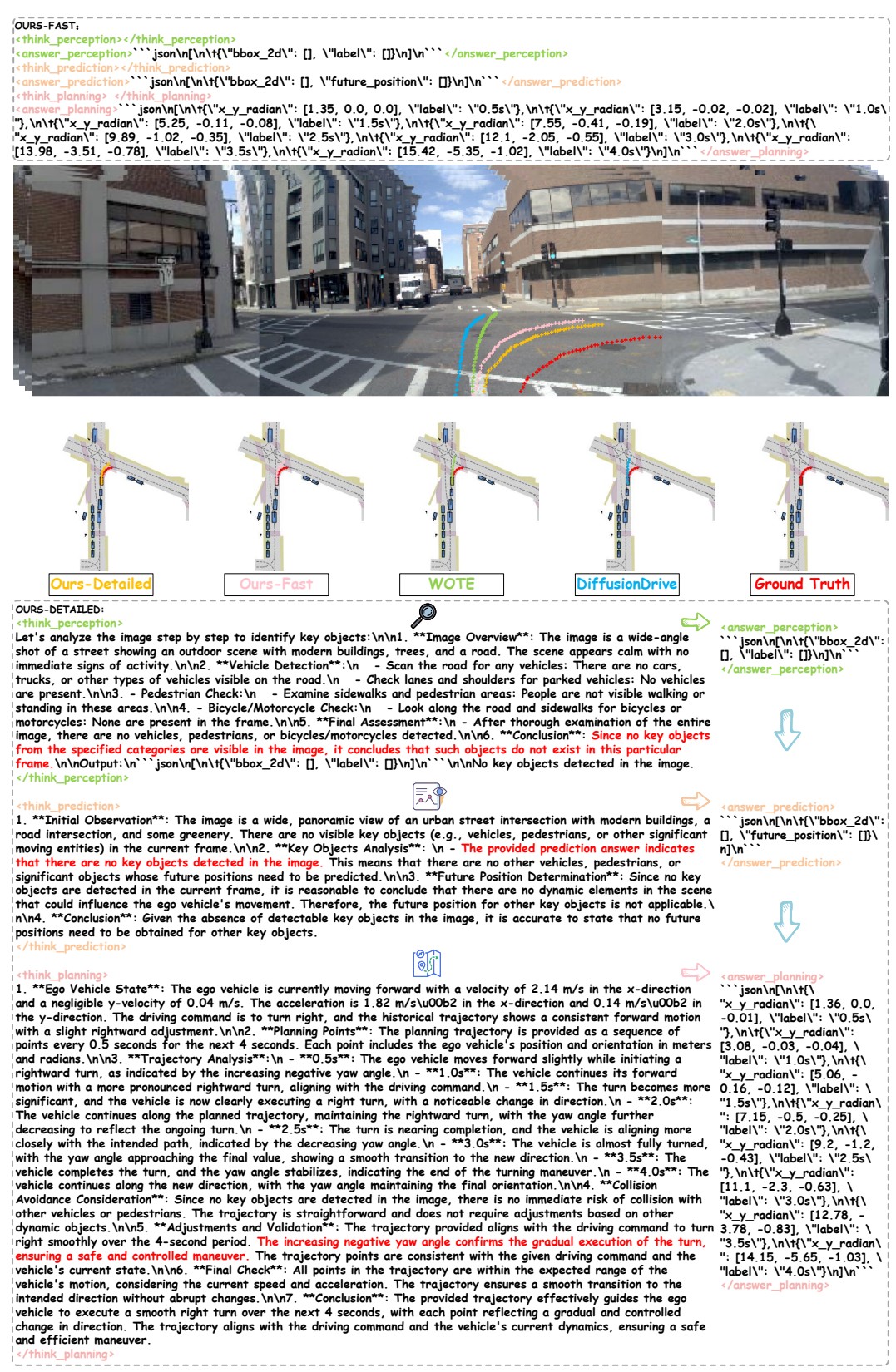

Figure 17: **Visualization NAVSIM cases with completed questions and answers.** *AutoDrive-P³* successfully recognizes the "Turn Right" command and provide the right planning trajectory, while other methods drive towards the building leading to collisions.

## G PROMPTS

In this section, we provide the completed and specific prompts used in the training/inference and prompts used to generate $P^3$-$CoT$ dataset. The prompt used in the training/inference is as follows.

```
You are an expert driving assistant. As an expert driving assistant,
    analyze the 3-second driving video context and answer the perception,
     prediction and planning question in the final frame.
Output format is '<think_perception> </think_perception>\n<
    answer_perception> </answer_perception>\n<think_prediction> </
    think_prediction>\n<answer_prediction> </answer_prediction>\n<
    think_planning> </think_planning>\n<answer_planning> </
    answer_planning>'.
Output the step-by-step Chain-of-Thought (CoT) reasoning process in <
    think> </think> tags and final answer in <answer> </answer> tags,
    respectively.
Ego Future Action is [Ego_Future_Action]. You current vehicle speed is [
    VEHICLE_SPEED] m/s, and the historical trajectory of the ego vehicle
    is [HISTORICAL_TRAJECTORY].
```

To sufficiently extract knowledge from Qwen2.5-VL-72B, we ask Qwen2.5-VL-72B to output the Chain-of-Thought (CoT) step by step. Qwen2.5-VL-72B is also required to use the CoT of perception tasks when it generates the CoT of prediction, and use the CoT of perception and prediction tasks when it generates the CoT of planning.

```
# Perception CoT
PROMPT_FORMAT = """I will provide you with a final frame image of video,
    an original question, and its answer related to the image. Your task
    is to answer it requires step-by-step Chain-of-Thought (CoT)
    reasoning with numerical or mathematical expressions where applicable
    . The reasoning process can include expressions like "let me think,"
    "oh, I see," or other natural language thought expressions.
Input Format:
Original Question: {original_question}
Original Answer: {original_answer}
Output Format:
<think>step-by-step reasoning process</think>
<answer>easy to verify answer</answer>
"""
QUESTION = "Examine the final frame image of video for key objects and
    report the coordinates of each detected object. Key object categories
     include: car, bus, truck, bicycle, pedestrian, motorcycle. The image
     size is 896x504."
ANSWER_FORMAT = "[\n\t{{\"bbox_2d\": {bbox}, \"label\": \"{label}\"}}\n]"

# Prediction and Planning
PROMPT_FORMAT = """I will provide you with a final frame image of video,
    the key objects in this frame, an question, a video Context, vehicle
    speed, historical trajectory (last 3 seconds) and its prediction and
    planning answer. Your task is to answer it requires step-by-step
    Chain-of-Thought (CoT) reasoning with numerical or mathematical
    expressions where applicable. The reasoning process can include
    expressions like "let me think," "oh, I see," or other natural
    language thought expressions.
Note that prediction and planning answers are the next 3-second future
    action for each object and ego vehicle planning trajectory. Video
    context is the 3-second context.
Input Format:
Key Objects: {key_objs}
Question: {original_question}
Video Context: {original_thinking}
Current Vehicle Speed: {vehicle_speed} m/s
Historical Trajectory (last 3 seconds, meters): {Historical_Trajectory}
Prediction Answer: \n{original_answer_predition}
```

```
Planning Answer: \n{original_answer_planning}
Output Format:
<think_prediction>step-by-step prediction reasoning process</
    think_prediction>
<answer_prediction>easy to verify prediction answer</answer_prediction>
<think_planning>step-by-step planning reasoning process</think_planning>
<answer_planning>easy to verify planning answer</answer_planning>
"""
QUESTION = """Predict the future action for each object and give the the
    ego vehicle planning trajectory. Future action can be: stop, straight
    , right, left. Planning trajectory is 6 points in the next 3 seconds
    (each point means 0.5s).
Please use the format as [x, y] in meters, where x-axis is perpendicular,
     and y-axis is parallel to the direction you are facing.
If y > 0, it means that the ego is to GO STRAIGHT, and vice yersa.
If x > 0, it means that the ego is to TURN RIGHT, and vice versa.
Note that current Vehicle Speed does affect the ego vehicle planning
    trajectory but you also should consider Historical Trajectory, Key
    Objects' Predicion Answers, Ego Action and the Video Context.
Uing numerical or mathematical expressions where applicable.
"""
```

