# OpenReview forum: "$AutoDrive\text{-}P^3$: Unified Chain of Perception–Prediction–Planning Thought via Reinforcement Fine-Tuning"
_ICLR.cc/2026/Conference — ICLR 2026 Poster_

### Official Review · Reviewer_K3hP · 2025-10-23

**Soundness:** 3
**Presentation:** 3
**Contribution:** 3
**Rating:** 6
**Confidence:** 3

**Summary:**

This paper proposes AutoDrive, a unified vision-language-action (VLA) model for autonomous driving. The key idea is to train a single model that can understand visual inputs, interpret instructions, and output low-level driving actions. The authors introduce a new training method called P³-GRPO (Perception–Planning–Policy Generalized Reinforcement Preference Optimization), which combines supervision from perception, planning, and control tasks in a joint optimization framework.

The model is evaluated on standard driving benchmarks and shows strong performance across all metrics. The collision rate in particular is much lower than prior systems like DriveVLM and DriveGPT, and the authors demonstrate that the proposed P³-GRPO method is the main driver behind these improvements.

**Strengths:**

The experimental results are really strong. The improvement in success rate and collision reduction is quite large compared to previous SOTA, which is impressive.

The P³-GRPO training framework seems like a solid contribution. It’s a clean idea that unifies different supervision signals under one preference optimization scheme, and it clearly helps the model learn better driving behaviors.

The paper is well written and easy to follow, with a clear structure and convincing ablations showing the effect of P³-GRPO.

**Weaknesses:**

The biggest missing piece is runtime analysis. There’s no mention of latency or inference speed anywhere in the paper. For a VLM-based driving model, this is a serious omission. These models are large and autoregressive, and it’s hard to tell whether the proposed system can actually run in real time.

There’s also no discussion of how the model could be deployed in practice. Prior work like DriveVLM already acknowledged this issue and proposed a hybrid setup: a lightweight driving system for high-frequency control and a heavy VLM that only runs in rare or complex cases. That kind of hybrid design makes practical sense, but this paper doesn’t touch on it at all.

Overall, while the performance is strong, the work feels a bit detached from deployment realities. It would be much stronger if the authors at least analyzed compute requirements or discussed how AutoDrive could fit into a practical driving stack.

**Questions:**

1. What’s the actual runtime per frame or per driving decision? Without this, it’s hard to judge feasibility.

2. Could the system be applied in a hybrid setup (like DriveVLM’s), where the large model runs only in low-frequency cases?

---

> ### Author Response · Authors · 2025-11-21
> **Thanks and response to Reviewer K3hP (1/2)**
>
> Thank you for your valuable and insightful comments. We deeply appreciate your input and have made every effort to address your concerns, as detailed below.
>
> ## Response to W1 and Q1: Dual Thinking Modes
>
> Thank you for this valuable suggestion. We agree that a hybrid setup like DriveVLM's presents an interesting approach for balancing computational efficiency and reasoning capability. However, our current work focuses primarily on establishing the viability of fully integrated reasoning through the $P^3\text{-}CoT$ framework, where perception, prediction and planning are jointly optimized within a reasoning chain. The architectural paradigm of a hybrid system presents challenges for maintaining the coherent, end-to-end reasoning process that our $P^3\text{-}CoT$ methodology aims to achieve. **However, to address the important concern of computational efficiency across high/low frequency, we have implemented a "dual-thinking" framework within our unified architecture.** This approach provides both fast (answer-only) and detailed (full reasoning) modes, effectively serving as an alternative solution to handle varying frequency requirements without compromising our integrated reasoning framework.
>
> **Figure 1. Dual-thinking**
> ```
> ┌─────────────────────┐         ┌─────────────────────┐         ┌─────────────────────┐
> │      Perception     │         │      Prediction     │         │       Planning      │
> │ (Thinking + Answer) │  ---->  │ (Thinking + Answer) │  ---->  │ (Thinking + Answer) │
> └─────────────────────┘         └─────────────────────┘         └─────────────────────┘
>                          Ours-Detailed Thinking (With Modules’ CoT)
>
> ┌─────────────────────┐         ┌─────────────────────┐         ┌─────────────────────┐
> │      Perception     │         │      Prediction     │         │       Planning      │
> │    (Only Answer)    │  ---->  │    (Only Answer)    │  ---->  │    (Only Answer)    │
> └─────────────────────┘         └─────────────────────┘         └─────────────────────┘
>                           Ours-Fast Thinking (Without Modules’ CoT)
> ```
>
>
> We have added comprehensive results and discussion of this dual-thinking approach to the revised manuscript (Tables 1-3 of updated manuscript). **Importantly, even our fast-thinking mode achieves SOTA performance across all benchmarks, while the detailed-thinking mode gets better planning performance.** This demonstrates that our architecture effectively balances performance and efficiency, providing a flexible solution for different operational scenarios. The dual thinking modes has been included in the revised manuscript. Main result as follow:
>
> **Table 1. Results on nuScenes Benchmark**
> |Method| L2(m) 1s$\downarrow$ | L2(m) 2s$\downarrow$ | L2(m) 3s$\downarrow$ | L2 (m) Avg.$\downarrow$ | Collision(%) 1s$\downarrow$ | Collision(%) 2s$\downarrow$ | Collision(%) 3s$\downarrow$  | Collision(%) Avg.$\downarrow$ |VLM|
> |-|:-:|:-:|:-:|:-:|:-:|:-:|:-:|:-:|:-:|
> | UniAD (23’ CVPR) |0.44|0.67|0.96|0.69|0.04|0.08|0.23|0.12|-|
> | AutoVLA (25’ NeurIPS) |0.25|0.46|0.73|0.48|0.07|0.07|0.26|0.13| Qwen2.5-VL-3B |
> | $AutoDrive\text{-}P^3$ (Ours-Detailed)    |0.15|0.30|0.54|**0.33**|0.00|0.02|0.15|**0.06**| Qwen2.5-VL-3B |
> | $AutoDrive\text{-}P^3$ (Ours-Fast)        |0.16|0.31|0.56|  0.34  |0.00| 0.04|0.20|  0.08  | Qwen2.5-VL-3B |
>
> **Table 2. Results on NAVSIMv1 Benchmark**
> |Method| NC$\uparrow$ | DAC$\uparrow$ | EP$\uparrow$ | TTC$\uparrow$ | Comf$\uparrow$ | PDMS$\uparrow$ |
> |-|:-:|:-:|:-:|:-:|:-:|:-:|
> | DiffusionDrive (25’ CVPR) |98.2|96.2|82.2|94.7|100.0|88.1|
> | WoTE (25’ ICCV) |98.5|96.8|81.9|94.9|99.9|88.3|
> | $AutoDrive\text{-}P^3$ (Ours-Detailed) |99.1|97.4|84.8|96.5|100.0|**90.6** |
> | $AutoDrive\text{-}P^3$ (Ours-Fast) |98.9|97.7|83.7|96.6|99.9|  90.2  |
>
> **Table 3. Results on NAVSIMv2 Benchmark**
> |Method| NC$\uparrow$ | DAC$\uparrow$ | DDC$\uparrow$ | TLC$\uparrow$ | EP$\uparrow$ | TTC$\uparrow$ | LK$\uparrow$ | HC$\uparrow$ | EC$\uparrow$ | EPDMS$\uparrow$  False/True |
> |-|:-:|:-:|:-:|:-:|:-:|:-:|:-:|:-:|:-:|:-:|
> | DiffusionDrive (25’ CVPR) |98.2|96.2|99.5|99.8|87.4|97.3|96.9|98.4|87.7| 84.7 / 88.2 |
> | WoTE (25’ ICCV)  |98.5|96.8|98.8|99.8|86.1|97.9|95.5|98.3|82.9| 84.2 / 87.7 |
> | $AutoDrive\text{-}P^3$ (Ours-Detailed)  |99.1|97.4|99.2|99.8|88.0|98.7|96.3|98.3|85.5|**86.2** / **89.9**|
> | $AutoDrive\text{-}P^3$ (Ours-Fast) |98.9|97.6|98.9|99.8|86.8|98.5|95.4|98.3|80.6| 85.2 / 88.7 |

---

> ### Author Response · Authors · 2025-11-21
> **Thanks and response to Reviewer K3hP (2/2)**
>
> ## Response to W2 \& Q2: Runtime Analysis
>
> As shown in **Figure 1. Dual-thinking** above (the same as Fig. 5 of updated manuscript), we implement a dual-thinking setup consisting of fast and detailed versions. While both adhere to the $P^3\text{-}CoT$ structure, the fast version outputs only final answers per module without reasoning, whereas the detailed version provides complete reasoning chains alongside answers. Runtime performance is as follow. **Using FlashAttention-2 and vLLM 0.8.0 on an A100 GPU, our fast version achieves 1.0 Hz, matching the inference speed of AutoVLA-Fast (NeurIPS' 25).** Importantly, our method demonstrates significantly better performance in both L2 distance (0.34 m vs. 0.48 m) and collision rate (0.08\% vs. 0.13\%). Despite this efficiency-performance balance, we acknowledge that computational complexity remains a key challenge, and improving real-time inference will be an important focus of our future work. The runtime analysis has been included in the revised manuscript. Main result as follow:
>
> | Method                 | Avg. L2 (m) $\downarrow$ | Avg. Collision (%) $\downarrow$ | FPS (Hz) $\uparrow$ |
> |------------------------|:---------------------------:|:----------------------------------:|:---------------------:|
> | UniAD (23' CVPR)       | 0.69                      | 0.12                             | 1.8                 |
> | DriveVLM (24' CoRL)    | 0.40                      | 0.27                             | **2.4**                 |
> | AutoVLA (Slow / Fast, 25' NeurIPS)  | 0.48 / 0.43                    | 0.13 / 0.19                             | 0.1 / 0.9         |
> | Ours (Detailed / Fast) | **0.33** / 0.34               | **0.06** / 0.08                      | 0.5 / 1.0          |

---

### Official Review · Reviewer_DPeP · 2025-10-29

**Soundness:** 3
**Presentation:** 3
**Contribution:** 3
**Rating:** 8
**Confidence:** 4

**Summary:**

This paper introduces AutoDrive-P³, a VLM framework for autonomous driving that unifies perception, prediction, and planning through structured Chain-of-Thought (CoT) reasoning. It proposes the P³-CoT dataset and P³-GRPO, a hierarchical reinforcement learning method to supervise all three stages, aiming for improved synergy and planning performance.

**Strengths:**

The paper's significance lies in its explicit attempt to unify perception, prediction, and planning within a VLM using CoT, addressing the limitations of fragmented or planning-only approaches. The quality is demonstrated through the creation of the P³-CoT dataset, designed to facilitate this unified reasoning , and the novel P³-GRPO algorithm, which extends reinforcement learning supervision beyond just planning. The clarity is good, effectively outlining the problem and the proposed solution. The strong results on nuScenes and NAVSIM suggest the approach improves planning reliability and interpretability.

**Weaknesses:**

The reliance on Qwen2.5-VL-72B to generate the P³-CoT dataset introduces potential biases or limitations inherent in the generating model. While manually verified, the quality of the generated CoT is crucial and may propagate errors. Additionally, the hierarchical nature of P³-GRPO, while intended to create synergy, could potentially lead to error accumulation if early stages (perception, prediction) are poorly optimized, impacting the final planning output. The paper acknowledges hallucination issues and the need for real-world testing, which are important next steps.

**Questions:**

1. The P³-CoT dataset was generated using Qwen2.5-VL-72B. How was the quality and correctness of the generated CoT reasoning verified beyond just checking the final labels? Could biases from the generating model affect the downstream training, especially on a different series of models?
2. P³-GRPO applies hierarchical supervision. Could errors or suboptimal performance in the earlier perception and prediction reward stages negatively impact the final planning stage more than a planning-only reward might? How sensitive is the final planning performance to the accuracy of the perception/prediction rewards?
3. The reward weights for P³-GRPO were set in a 1:2:2:5 ratio (format:perception:prediction:planning). Could you elaborate on the process for determining these weights? Have other weighting schemes been explored?
4. The paper argues that fragmented VLM approaches lack synergy. Table 3 compares SFT-only, Planning-GRPO, and P³-GRPO. While P³-GRPO performs best, the Planning-GRPO version also shows improvement over SFT-only. Could you further discuss the specific benefits attributed solely to the hierarchical supervision (P³-GRPO) compared to just applying GRPO to planning?

---

> ### Author Response · Authors · 2025-11-21
> **Thanks and response to Reviewer DPeP (1/3)**
>
> Thank you for your valuable and insightful comments. We deeply appreciate your input and have made every effort to address your concerns, as detailed below.
>
> ## Response to W1 \& Q1: Different Generative Model and Validation the CoT Dataset
>
> We have compared the quality and training performance of ${P^3\text{-}CoT}$ dataset generated separately by GPT4o and Qwen2.5-VL-72B on nuScenes. **We conduct same training setting and there is almost no difference.** Considering that Qwen2.5-VL-72B is open source, we use Qwen2.5-VL-72B to generate ${P^3\text{-}CoT}$ of NAVSIM with larger data amount.
>
> To guarantee the quality of the ${P^3\text{-}CoT}$ dataset, we adopte a assessment following DriveLM [1]. At the holistic level, each CoT label is manually inspected to ensure it strictly follows the prescribed reasoning structure—progressing completely and sequentially from perception to prediction and then to planning, with all final module outputs present and correctly formatted. At the modular level, we perform a manual check for factual consistency and reasoning quality. The dataset is divided into 10 splits, each assigned to three independent annotators. They randomly sample 10\% of their assigned split and meticulously examine the alignment between the generated CoT and the ground-truth labels for each module. This includes assessing the factual consistency of the reasoning steps, the fluency of the language, and the overall logical plausibility. In both levels of inspection, each reviewed CoT is labeled as "good" or "poor". The accuracy for a split is calculated based on the proportion of "good" samples. We set a passing accuracy threshold of 90\% for a split to be considered qualified. Any split failing to meet this standard is deemed imperfect, and the CoT data for that portion is regenerated and resubmitted for assessment. This iterative process continues until all splits pass the manual quality check by all annotators.
>
> The accuracy and effectiveness of our $P^3\text{-}CoT$ dataset receive strong support from two key experimental results.
>
> **First**, the performance advantage of the full $P^3\text{-}CoT$ over the "only planning" baseline (Table 4, line 5 vs. line 4 of updated manuscript) validates the effectiveness of the holistic CoT structure. Main result as follow:
>
> **Table 1. Only Planning CoT or $P^3\text{-}CoT$**
> | Method | Perception$\uparrow$ | Prediction$\uparrow$ | Planning (Collision,%) 1s$\downarrow$ | Planning (Collision,%) 2s$\downarrow$| Planning (Collision,%) 3s$\downarrow$|Planning (Collision,%) Avg.$\downarrow$ |
> |-|:-:|:-:|:-:|:-:|:-:|:-:|
> | SFT + Only Planning CoT GRPO | -- | -- |0.03|0.08|0.24|0.1|
> | SFT + $P^3\text{-}CoT$ GRPO |**0.64**|**0.54**|**0.00**|**0.02**|**0.15**|**0.06**|
>
> **Second**, the superior performance of the slow-thinking mode (with full CoT) compared to the fast-thinking mode (without reasoning chains) across multiple benchmarks (Tables 1-3 of updated manuscript) confirms the value of accurate and detailed granular CoT reasoning. Together, these results provide comprehensive validation for both the overall framework and the detailed reasoning steps in our $P^3\text{-}CoT$ dataset. We have revised the manuscript and added relevant validation of $P^3\text{-}CoT$ dataset at Appendix B.5. Main result as follow:
>
> **Table 2. $P^3\text{-}CoT$ on nuScenes**
> |Method| L2(m) 1s$\downarrow$ | L2(m) 2s$\downarrow$ | L2(m) 3s$\downarrow$ | L2 (m) Avg.$\downarrow$ | Collision(%) 1s$\downarrow$ | Collision(%) 2s$\downarrow$ | Collision(%) 3s$\downarrow$  | Collision(%) Avg.$\downarrow$ |VLM|
> |-|:-:|:-:|:-:|:-:|:-:|:-:|:-:|:-:|:-:|
> | Ours-Detailed (With Modules’ CoT)    |0.15|0.30|0.54|**0.33**|0.00|0.02|0.15|**0.06**| Qwen2.5-VL-3B |
> | Ours-Fast (Without Modules’ CoT)        |0.16|0.31|0.56|  0.34  |0.00| 0.04|0.20|  0.08  | Qwen2.5-VL-3B |
>
> **Table 3. $P^3\text{-}CoT$ on NAVSIMv1**
> |Method| NC$\uparrow$ | DAC$\uparrow$ | EP$\uparrow$ | TTC$\uparrow$ | Comf$\uparrow$ | PDMS$\uparrow$ |
> |-|:-:|:-:|:-:|:-:|:-:|:-:|
> | Ours-Detailed (With Modules’ CoT) |99.1|97.4|84.8|96.5|100.0|**90.6** |
> | Ours-Fast (Without Modules’ CoT) |98.9|97.7|83.7|96.6|99.9|  90.2  |
>
> **Table 4. $P^3\text{-}CoT$ on NAVSIMv2**
> |Method| NC$\uparrow$ | DAC$\uparrow$ | DDC$\uparrow$ | TLC$\uparrow$ | EP$\uparrow$ | TTC$\uparrow$ | LK$\uparrow$ | HC$\uparrow$ | EC$\uparrow$ | EPDMS$\uparrow$  False/True |
> |-|:-:|:-:|:-:|:-:|:-:|:-:|:-:|:-:|:-:|:-:|
> | Ours-Detailed (With Modules’ CoT)  |99.1|97.4|99.2|99.8|88.0|98.7|96.3|98.3|85.5|**86.2** / **89.9**|
> | Ours-Fast (Without Modules’ CoT) |98.9|97.6|98.9|99.8|86.8|98.5|95.4|98.3|80.6| 85.2 / 88.7 |
>
> [1] DriveLM: Driving with Graph Visual Question Answering. (24’ ECCV)

---

> ### Author Response · Authors · 2025-11-21
> **Thanks and response to Reviewer DPeP (2/3)**
>
> ## Response to W2 \& Q2: Hierarchical Supervision
>
> We thank the reviewer for this insightful question. Indeed, **the inherent interdependence among perception, prediction, and planning modules means that suboptimal rewards in earlier stages can initially impact planning performance**. This sensitivity is an inherent characteristic of any coherent reasoning system where modules build upon one another. As shown in Fig. 8 of updated manuscript, while planning performance is initially poorer when perception and prediction rewards are low, our ${P^3\text{-}GRPO}$ framework overcomes this through joint optimization. The rewards for all three modules rise synergistically during training, with improvements in perception and prediction directly enabling the planning module to eventually achieve SOTA performance.
>
>
>
> ## Response to Q3: Ablation Study on Reward Weight
>
> While outstanding planning performance is the ultimate goal of autonomous driving systems, we raise the weight of planning reward properly in our setting (format : perception : prediction : planning = 1:2:2:5). Furthermore, we also test the 1:1:1:7 reward weight, and the results are shown in Table 10 of updated manuscript. Overemphasizing the planning reward can hinder the optimization of perception and prediction modules. This imbalance ultimately leads to inferior overall planning performance, as accurate planning is contingent upon reliable inputs from the preceding stages. Therefore, we finally choose the weights as 1:2:2:5. As suggested, we have added this experiment along with its full results to Appendix C of our revised manuscript. Main result as follow:
>
> | Reward weight | Perception $\uparrow$ | Prediction $\uparrow$ | Planning (Avg. L2) $\downarrow$ |
> |:---------------:|:-----------------------:|:-----------------------:|:---------------------------------:|
> | 1:2:2:5       | **0.64**              | **0.54**              | **0.33**                        |
> | 1:1:1:7       | 0.63                  | 0.53                  | 0.34                            |

---

> ### Author Response · Authors · 2025-11-21
> **Thanks and response to Reviewer DPeP (3/3)**
>
> ## Response to Q4: Further Discuss for ${P^3\text{-}GRPO}$
>
> The performance improvement of Planning-GRPO over the SFT-only baseline can indeed be attributed to the reinforcement learning process that directly optimizes planning decisions. However, **since Planning-GRPO supervises only the final planning output, it lacks the mechanism to detect or correct inaccuracies in upstream perception and prediction modules**. This limitation fundamentally constrains its potential, as errors in these foundational stages inevitably propagate to and impair the final planning quality.
>
> This motivates us to propose the ${P^3\text{-}GRPO}$ framework, which introduces hierarchical reward signals to jointly optimize all three modules within the P³-CoT structure. By reinforcing perceptual grounding, trajectory forecasting, and final planning in a coordinated manner, we ensure that the model builds its decisions on a consistent and accurate understanding. This synergistic optimization enables ${P^3\text{-}GRPO}$ to achieve significantly higher performance and lower collision rates than Planning-GRPO, as conclusively demonstrated in Table 4 of updated manuscript. Main result as follow:
>
> | Method                                            | Perception$\uparrow$ | Prediction$\uparrow$ | Planning (Collision,%) 1s$\downarrow$ | Planning (Collision,%) 2s$\downarrow$| Planning (Collision,%) 3s$\downarrow$|Planning (Collision,%) Avg.$\downarrow$ |
> |---------------------------------------------------|:------------:|:------------:|:------------------------------:|:----------:|:------:|:------:|
> | Only SFT                 | 0.33       | 0.23       | 0.01                    | 0.08 | 0.40 | 0.17 |
> | SFT + Only Planning GRPO | --         | --         | 0.03                    | 0.08 | 0.24 | 0.12 |
> | SFT + $P^3\text{-}GRPO$  | **0.64**       | **0.54**       | **0.00**                    | **0.02** | **0.15** | **0.06** |

---

### Official Review · Reviewer_NDzW · 2025-11-01

**Soundness:** 3
**Presentation:** 3
**Contribution:** 2
**Rating:** 4
**Confidence:** 5

**Summary:**

This paper presents AutoDrive-P3, a unified end-to-end autonomous driving framework that integrates Perception, Prediction, and Planning through structured chain-of-thought reasoning. It introduces a new P3-CoT dataset and a hierarchical reinforcement learning algorithm (P3-GRPO) to ensure coherent reasoning across tasks.

**Strengths:**

1. The paper is easy to read, and the core idea is clearly presented.

**Weaknesses:**

1. As mentioned in the Introduction, both perception and prediction are important for planning, which I agree with. However, as described in the Method section, there is no direct connection from perception to planning, which seems inconsistent with Figure 1. I wonder how the information or features are transferred from perception to planning. How does the perception-CoT contribute to the planning-CoT?

2. The authors should include a Related Works subsection to discuss prior work on VLM-based autonomous driving. There are many studies that use VLMs to enhance end-to-end autonomous driving across all three tasks. Moreover, different strategies exist for integrating VLMs into the pipeline. For example, [1] encodes the VLM into the pipeline and jointly fine-tunes it, while [2] uses the VLM to generate an auxiliary dataset to improve planning, a strategy that is similar to the proposed dataset generation approach.

[1] J. Hwang et al., “EMMA: End-to-End Multimodal Model for Autonomous Driving,” TMLR, 2025.

[2] Y. Xu et al., “VLM-AD: End-to-End Autonomous Driving through Vision-Language Model Supervision,” CoRL, 2025

3. The results on nuScenes are close to those of OmniDrive and OpenDriveVLA in L2. Some discussion on possible reasons for this would be helpful, especially since this work emphasizes the benefits of jointly considering all three tasks.

**Questions:**

1. In Section 4.1, the data labeling pipeline is introduced, with more details in the Appendix. I am curious about what the prompt questions look like for the three tasks. In many related works, task-specific questions are carefully designed to extract knowledge from VLMs, what is the strategy here?

2. How is the quality of the P^{3}-COT dataset ensured? Since the labels are generated by Qwen2.5-VL-72B, did the authors encounter imperfect labels during annotation? How was label quality evaluated? Were any quality-control methods such as cross-validation or human evaluation (e.g., questionnaires) applied?

---

> ### Author Response · Authors · 2025-11-21
> **Thanks and response to Reviewer NDzW (1/3)**
>
> Thank you for your valuable and insightful comments. We deeply appreciate your input and have made every effort to address your concerns, as detailed below.
>
> ## Response to W1: How ${P^3\text{-}CoT}$ works?
>
> We thank the reviewer for this insightful question regarding the flow of information from perception to planning, which is central to our framework. **The connection of ${P^3\text{-}CoT}$ is not made through a separate architectural component, but is explicitly and causally enforced through the sequential, structured natural language of the CoT itself.**
>
> In our ${P^3\text{-}CoT}$ paradigm, the model is required to generate the reasoning steps in a strict sequence: **Perception → Prediction → Planning**. The output of each module is produced as structured text, which directly serves as the premise for the next module. Specifically: Perception-CoT identifies and describes key entities. Prediction-CoT then takes these perceived entities as its subject, forecasting their future dynamics based on the perceptual input. Finally, Planning-CoT synthesizes the outputs from both previous modules to make a safe decision.
>
> As shown in Fig. 6(a) of updated manuscript, Perception-CoT identifies a pedestrian, and Prediction-CoT predict it will leave the current driving area. It is this combined understanding that allows the Planning-CoT to logically conclude: "Since the pedestrian does not affect the vehicle, the ego vehicle can start moving." As the comparison in the figure shows, this leads to a correct and timely "Move" decision, whereas other methods, lacking this explicit reasoning chain, remain "Stop".
>
> Our ablation experiments in Table 4 of updated manuscript also provide the results comparison between only Planning-CoT (line 4) and ${P^3\text{-}CoT}$ (line 5). The results show that ${P^3\text{-}CoT}$ can improve planning performance greatly compared to supervision only for the planning, which indicates the way of Perception-CoT contribution to Planning-CoT. Main result as follow:
>
> **Table 1. Only Planning or $P^3\text{-}CoT$**
> | Method | Perception$\uparrow$ | Prediction$\uparrow$ | Planning (Collision,%) 1s$\downarrow$ | Planning (Collision,%) 2s$\downarrow$| Planning (Collision,%) 3s$\downarrow$|Planning (Collision,%) Avg.$\downarrow$ |
> |-|:-:|:-:|:-:|:-:|:-:|:-:|
> | SFT + Only Planning GRPO | -- | -- |0.03|0.08|0.24|0.1|
> | SFT + $P^3\text{-}CoT$ GRPO |**0.64**|**0.54**|**0.00**|**0.02**|**0.15**|**0.06**|
>
>
>
>
> ## Response to W2: Other VLM-based Methods
>
> Thank you for your valuable suggestions on our writing. (1) Following your advice, we reorgnize our related work section and add subsection 2.2 "VLM-based Autonomous Driving Methods" to review prior work on VLM-based autonomous driving. We have introduced EMMA before and add more related work, including VLM-AD. (2) Though existing VLM-based AD studies try to enhance the planning performance through seperate perception, prediction, and planing, they lack joint optimization of all three modules, as shown in Fig. 1(c). Our method aims to jointly optimize perception, prediction and planning modules to achieve better planning result, which is emphasized in related work.
>
> The corresponding sentences about references are as follows: "Approaches including DriveVLM, EMMA, VLM-AD, OpenEMMA, OmniDrive, OpenDriveVLA, and AutoVLA benefit from VLMs’ rich world knowledge and reasoning capabilities, demonstrating strong performance in driving scenarios.".

---

> ### Author Response · Authors · 2025-11-21
> **Thanks and response to Reviewer NDzW (2/3)**
>
> ## Response to W3: L2 Disscusion
>
> We thank the reviewer for raising this point regarding the comparable L2 performance on nuScenes. In our opinion, **L2 error alone, especially when values are close among models, may not fully reflect the driving capability of an AD system**. Our response is twofold:
>
> First, while our method achieves competitive L2 results, **its primary advantage lies in significantly enhancing driving safety with the same L2 performance**. As shown in Table 1 of updated manuscript, our ${P^3\text{-}GRPO}$ framework, which jointly optimizes perception, prediction, and planning, reduces the collision rate by approximately 40\% compared to SOTA methods under similar SOTA L2 performance. This demonstrates that our approach better balances trajectory accuracy with safety-critical decision-making. Main result as follow:
>
> **Table 2. Results on nuScenes Benchmark**
> |Method| L2(m) 1s$\downarrow$|L2(m) 2s$\downarrow$|L2(m) 3s$\downarrow$|L2 (m) Avg.$\downarrow$ |Collision(%) 1s$\downarrow$|Collision(%) 2s$\downarrow$|Collision(%) 3s$\downarrow$|Collision(%) Avg.$\downarrow$|VLM|
> |-|:-:|:-:|:-:|:-:|:-:|:-:|:-:|:-:|:-:|
> | UniAD (23’ CVPR) |0.44|0.67|0.96|0.69|0.04|0.08|0.23|0.12|-|
> | AutoVLA (25’ NeurIPS) |0.25|0.46|0.73|0.48|0.07|0.07|0.26|0.13|Qwen2.5-VL-3B|
> | Ours-Detailed (With Modules’ CoT)|0.15|0.30|0.54|**0.33**|0.00|0.02|0.15|**0.06**|Qwen2.5-VL-3B|
> | Ours-Fast (Without Modules’ CoT)|0.16|0.31|0.56|0.34|0.00| 0.04|0.20|0.08|Qwen2.5-VL-3B|
>
> Second, as supported by prior work [1], L2 distance error (or ADE) in open-loop settings may not reliably correlate with closed-loop driving performance:
>
> >**"However, existing metrics such as the average displacement error (ADE) between a predicted and recorded human trajectory often misrepresent the relative accuracy of trajectories."** (NAVSIM, 24' NeurIPS)
>
> When L2 error is sufficiently low, metrics such as collision rate and closed-loop evaluation become more meaningful indicators of a safe and effective AD system. To further validate the benefits of our joint reasoning strategy, we conducted extensive closed-loop experiments on the NAVSIMv1/v2 benchmark. As shown in Tables 2 and 3 of updated manuscript, **our method achieves SOTA performance on both NAVSIMv1/v2 (achieving 90.6 PDMS and 89.9 EPDMS), underscoring the practical advantages of our integrated approach in realistic driving scenarios**.
>
> **Table 3. Results on NAVSIMv1 Benchmark**
> |Method| NC$\uparrow$ | DAC$\uparrow$ | EP$\uparrow$ | TTC$\uparrow$ | Comf$\uparrow$ | PDMS$\uparrow$ |
> |-|:-:|:-:|:-:|:-:|:-:|:-:|
> | DiffusionDrive (25’ CVPR) |98.2|96.2|82.2|94.7|100.0|88.1|
> | WoTE (25’ ICCV) |98.5|96.8|81.9|94.9|99.9|88.3|
> | Ours-Detailed (With Modules’ CoT) |99.1|97.4|84.8|96.5|100.0|**90.6** |
> | Ours-Fast (Without Modules’ CoT) |98.9|97.7|83.7|96.6|99.9|  90.2  |
>
> **Table 4. Results on NAVSIMv2 Benchmark**
> |Method| NC$\uparrow$ | DAC$\uparrow$ | DDC$\uparrow$ |TLC$\uparrow$| EP$\uparrow$ | TTC$\uparrow$ | LK$\uparrow$ | HC$\uparrow$ | EC$\uparrow$ |EPDMS$\uparrow$  False/True|
> |-|:-:|:-:|:-:|:-:|:-:|:-:|:-:|:-:|:-:|:-:|
> | DiffusionDrive (25’ CVPR)|98.2|96.2|99.5|99.8|87.4|97.3|96.9|98.4|87.7|84.7 / 88.2|
> | WoTE (25’ ICCV)|98.5|96.8|98.8|99.8|86.1|97.9|95.5|98.3|82.9|84.2 / 87.7|
> | Ours-Detailed (With Modules’ CoT)|99.1|97.4|99.2|99.8|88.0|98.7|96.3|98.3|85.5|**86.2** / **89.9**|
> | Ours-Fast (Without Modules’ CoT)|98.9|97.6|98.9|99.8|86.8|98.5|95.4|98.3|80.6|85.2 / 88.7|
>
> [1] NAVSIM: Data-Driven Non-Reactive Autonomous Vehicle Simulation and Benchmarking. (24’ NeurIPS)
>
> ## Response to Q1: Prompt Design
>
> We provide the complete prompt templates in **Appendix F**. Our core strategy involves presenting the ground-truth answers to Qwen2.5-VL-72B and **instructing it to generate the corresponding reasoning steps step-by-step**. The model is explicitly required to utilize the Perception-CoT results when generating Prediction-CoT, and to build upon both Perception-CoT and Prediction-CoT when producing the final Planning-CoT, thereby ensuring a logically coherent reasoning chain.
>
> The primary criterion for a successful prompt is that it elicits a CoT output that correctly follows the prescribed $P^3$-CoT format. All generated reasoning chains undergo rigorous manual inspection following the quality assurance protocol detailed in our response to **Q2: Validation the CoT Dataset**.
>
> Regarding model selection, we have compared the quality and training performance of the $P^3$-CoT datasets generated by GPT-4o and Qwen2.5-VL-72B on the nuScenes benchmark. Under the same training settings, the resulting performance showed almost no difference. Given the open-source nature of Qwen2.5-VL-72B, we selecte it to generate the larger-scale $P^3$-CoT dataset for NAVSIM. We have added this clarification and the detailed prompts to **Appendix F** of the revised manuscript.

---

> ### Author Response · Authors · 2025-11-21
> **Thanks and response to Reviewer NDzW (3/3)**
>
> ## Response to Q2: Validation the CoT Dataset
>
> Our CoT dataset is structured at both holistic and modular levels, both critical to the model's performance. **The holistic CoT refers to the label of perception-prediction-planning within the ${P^3\text{-}CoT}$, while the modular CoT ensures the accuracy and consistency of each individual module's reasoning with respect to the video and its labels**.
>
> To guarantee the quality of the ${P^3\text{-}CoT}$ dataset, we adopte a assessment following DriveLM [5]. At the holistic level, each CoT label is manually inspected to ensure it strictly follows the prescribed reasoning structure—progressing completely and sequentially from perception to prediction and then to planning, with all final module outputs present and correctly formatted. At the modular level, we perform a manual check for factual consistency and reasoning quality. The dataset is divided into 10 splits, each assigned to three independent annotators. They randomly sample 10\% of their assigned split and meticulously examine the alignment between the generated CoT and the ground-truth labels for each module. This includes assessing the factual consistency of the reasoning steps, the fluency of the language, and the overall logical plausibility. In both levels of inspection, each reviewed CoT is labeled as "good" or "poor". The accuracy for a split is calculated based on the proportion of "good" samples. We set a passing accuracy threshold of 90\% for a split to be considered qualified. Any split failing to meet this standard is deemed imperfect, and the CoT data for that portion is regenerated and resubmitted for assessment. This iterative process continues until all splits pass the manual quality check by all annotators.
>
> The accuracy and effectiveness of our $P^3\text{-}CoT$ dataset receive strong support from two key experimental results.
>
> **First**, the performance advantage of the full $P^3\text{-}CoT$ over the "only planning" baseline (Table 4, line 5 vs. line 4 of updated manuscript) validates the effectiveness of the holistic CoT structure. **Main results are the same as Table 1. Only Planning CoT or $P^3\text{-}CoT$**
>
>
> **Second**, the superior performance of the slow-thinking mode (with full CoT) compared to the fast-thinking mode (without reasoning chains) across multiple benchmarks (Tables 1-3 of updated manuscript) confirms the value of accurate and detailed granular CoT reasoning. Together, these results provide comprehensive validation for both the overall framework and the detailed reasoning steps in our $P^3\text{-}CoT$ dataset. We have revised the manuscript and added relevant validation of $P^3\text{-}CoT$ dataset at Appendix B.5. **The main results are the same as in Table 2. (Results on nuScenes Benchmark), Table 3. (Results on NAVSIMv1 Benchmark), and Table 4. (Results on NAVSIMv2 Benchmar) above**.

---

### Official Review · Reviewer_PCAX · 2025-11-01

**Soundness:** 2
**Presentation:** 4
**Contribution:** 3
**Rating:** 6
**Confidence:** 4

**Summary:**

They introduce a chain of thought dataset that combines reasoning in Planning, Prediction and Perception. They aim to reduce the error Propagation gap of classical Systems, by conditioning Output on different reasoning Levels. They also contribute a unified GRPO Training Approach that aims to balance the learning contribution by each part of the System (Perception, Prediction, Planning).

**Strengths:**

They contribute a unification in VLM reasoning by conditioning Planning on Perception and Prediction. They additionally introduce the hierarchical GRPO Training Extension that is interesting and potentially/based on their results useful.

**Weaknesses:**

They justify their work with a broad Claim, that is not backed by any citation or Explanation: VLM-based AD NEEDS CoT. This cannot be stated as a simple fact. Additionally, based on the text and figure 2 (Backed by figure 5, 10, 11, 12, 13, 14) it does not seem to be the case, that the model really works with Videos. Perception CoT seems to be non-temporal, but based on a single frame. While this could be for visualization purposes and Explanation, an additional Statement or Explanation would be helpful. Otherwise it is not really understandable wether the Pipeline really conditions on temporal reasoning (which seems to be claimed) or not.

**Questions:**

1. Could you Elaborate on why a key Limitation of VLM-based E2E Systems is "Lack of Chain-of-Thought"?
2. Could you clarify wether you are using temporal Video data your VLM conditions on, or you are aggregating Frames/ Just conditioning on one frame?
3. How did you validate the CoT Dataset gt?
4. Can you Showcase that CoT reduces the error Propagation and the gap reduction really comes from using CoT and not from foundation model Training/ higher capacity of the model generally ?

---

> ### Author Response · Authors · 2025-11-21
> **Thanks and response to Reviewer PCAX (1/4)**
>
> Thank you for your valuable and insightful comments. We deeply appreciate your input and have made every effort to address your concerns, as detailed below.
>
> ## Response to W1 \& Q1: VLM-based AD needs CoT
>
> We claim that VLMs highly need Chain of Thought (CoT) due to two considerations:
>
> (1) Existing studies [1, 2] have shown that CoTs enhance the capabilities of LLM/VLM in different downstream tasks. For autonomous driving (AD) task, DriveVLM [3], OmniDrive [4], CoT-Drive [5], and DriveCoT [6] show that **explicit CoTs compel the model to articulate its reasoning step-by-step before concluding**, which enforces a structured, transparent, and more reliable reasoning process, thereby directly reducing the risk of unfounded guesses and hallucinations. (However, in these systems, the CoTs are often isolated modules limiting reasoning synergy. This fundamental limitation directly motivates our proposed ${AutoDrive\text{-}P^3}$ framework.)
>
> (2) An additional ablation study on "dual-thinking" further confirms the necessity of explicit CoT reasoning in AD tasks. As figure follows (Fig. 5 of updated manuscript), fast-thinking mode, which produces only the final answer in the ${P^3\text{-}CoT}$ format, serves as the baseline against detailed-thinking mode, which produces thinking and answer. **The better performance of the detailed-thinking mode (With Modules’ CoT) across nuScenes and NVASIMv1/v2**, as shown in Table 1-3 of updated manuscript, underscores the critical role of CoT. We have revised the manuscript and added relevant citations to the literature on CoT. Main result as follow:
>
> **Figure 1. Dual-thinking**
> ```
> ┌─────────────────────┐         ┌─────────────────────┐         ┌─────────────────────┐
> │      Perception     │         │      Prediction     │         │       Planning      │
> │ (Thinking + Answer) │  ---->  │ (Thinking + Answer) │  ---->  │ (Thinking + Answer) │
> └─────────────────────┘         └─────────────────────┘         └─────────────────────┘
>                          Ours-Detailed Thinking (With Modules’ CoT)
>
> ┌─────────────────────┐         ┌─────────────────────┐         ┌─────────────────────┐
> │      Perception     │         │      Prediction     │         │       Planning      │
> │    (Only Answer)    │  ---->  │    (Only Answer)    │  ---->  │    (Only Answer)    │
> └─────────────────────┘         └─────────────────────┘         └─────────────────────┘
>                           Ours-Fast Thinking (Without Modules’ CoT)
> ```
>
> **Table 1. Dual-thinking on nuScenes**
> |Method| L2(m) 1s$\downarrow$ | L2(m) 2s$\downarrow$ | L2(m) 3s$\downarrow$ | L2 (m) Avg.$\downarrow$ | Collision(%) 1s$\downarrow$ | Collision(%) 2s$\downarrow$ | Collision(%) 3s$\downarrow$  | Collision(%) Avg.$\downarrow$ |VLM|
> |-|:-:|:-:|:-:|:-:|:-:|:-:|:-:|:-:|:-:|
> | Ours-Detailed (With Modules’ CoT)    |0.15|0.30|0.54|**0.33**|0.00|0.02|0.15|**0.06**| Qwen2.5-VL-3B |
> | Ours-Fast (Without Modules’ CoT)        |0.16|0.31|0.56|  0.34  |0.00| 0.04|0.20|  0.08  | Qwen2.5-VL-3B |
>
> **Table 2. Dual-thinking on NAVSIMv1**
> |Method| NC$\uparrow$ | DAC$\uparrow$ | EP$\uparrow$ | TTC$\uparrow$ | Comf$\uparrow$ | PDMS$\uparrow$ |
> |-|:-:|:-:|:-:|:-:|:-:|:-:|
> | Ours-Detailed (With Modules’ CoT) |99.1|97.4|84.8|96.5|100.0|**90.6** |
> | Ours-Fast (Without Modules’ CoT) |98.9|97.7|83.7|96.6|99.9|  90.2  |
>
> **Table 3. Dual-thinking on NAVSIMv2**
> |Method| NC$\uparrow$ | DAC$\uparrow$ | DDC$\uparrow$ | TLC$\uparrow$ | EP$\uparrow$ | TTC$\uparrow$ | LK$\uparrow$ | HC$\uparrow$ | EC$\uparrow$ | EPDMS$\uparrow$  False/True |
> |-|:-:|:-:|:-:|:-:|:-:|:-:|:-:|:-:|:-:|:-:|
> | Ours-Detailed (With Modules’ CoT)  |99.1|97.4|99.2|99.8|88.0|98.7|96.3|98.3|85.5|**86.2** / **89.9**|
> | Ours-Fast (Without Modules’ CoT) |98.9|97.6|98.9|99.8|86.8|98.5|95.4|98.3|80.6| 85.2 / 88.7 |
>
> [1] Chain-of-thought prompting elicits reasoning in large language model. (22' NeurIPS)
>
> [2] Visual-RFT: Visual Reinforcement Fine-Tuning. (25' ICCV)
>
> [3] DriveVLM: The Convergence of Autonomous Driving and Large Vision-Language Models. (24' CoRL)
>
> [4] Omnidrive: A holistic vision-language dataset for autonomous driving with counterfactual reasoning. (25' CVPR)
>
> [5] CoT-Drive: Efficient motion forecasting for autonomous driving with llms and chain-of-thought prompting (25' IEEE TAI)
>
> [6] DriveCoT: Integrating Chain-of-Thought Reasoning with End-to-End Driving (24' arXiv)

---

> > ### Comment · Reviewer_PCAX · 2025-11-27
> >
> > Thank you for the additional ablation study and citations. The dual-thinking experiment provides useful empirical evidence that CoT improves performance in your setting, and the cited works (DriveVLM, OmniDrive, CoT-Drive) represent relevant state-of-the-art.
> > However, I maintain that the claim remains somewhat overclaimed. The cited works demonstrate that CoT can improve VLM-based AD systems, but none provide systematic analysis of when or why CoT is beneficial versus alternative approaches (e.g., increased model capacity, different supervision strategies). Your own ablation shows consistent but modest improvements (e.g., PDMS 90.2→90.6), which supports 'CoT is helpful' rather than 'CoT is necessary.'
> > I suggest softening the framing from 'VLM-based AD needs CoT' to 'VLM-based AD benefits from CoT' — this would better align with the evidence provided. This is a minor concern and does not significantly affect my overall assessment of the paper's contributions. Generally, you have significantly improved on my concern here.

---

> ### Author Response · Authors · 2025-11-21
> **Thanks and response to Reviewer PCAX (2/4)**
>
> ## Response to W2 \& Q2: Video or Image
>
> We are sorry for our insufficient expression. We would like to clarify that **our model is designed to process video inputs**. Within the ${P^3\text{-}CoT}$ framework, the Perception-CoT is focused on identifying key objects in the final frame to streamline computation and minimize redundant processing. This critical perceptual output then serves as the basis for subsequent prediction and planning CoTs, which leverage the broader history of visual information from the entire video sequence. The effectiveness of this video-based approach is confirmed by our ablation studies (Table 5, line 3 and line 4 of updated manuscript), demonstrating that **utilizing temporal information from video leads to superior performance**. In the original manuscript, for clarity in visualization, all presented examples depict only the perceptual results of the final frame. We have revised the figures in the manuscript to prevent any potential misunderstanding and apologize for the initial confusion. Main result as follow:
>
> | Method            | Group Size | Sensor Type | L2 (m)$\downarrow$ 1s | L2 (m)$\downarrow$ 2s | L2 (m)$\downarrow$ 3s  |L2 (m)$\downarrow$ Avg.  | Collision(%)$\downarrow$ 1s | Collision(%)$\downarrow$ 2s | Collision(%)$\downarrow$ 3s | Collision(%)$\downarrow$ Avg. |
> |-------------------|:------------:|:-------------:|:-----------------------:|:---------------------------:|:------:|:--------:|:------------------------------:|:---------------------------:|:---------------------------:|:---------------------------:|
> | Ablation 3        | 8          | Image       | 0.16                  | 0.32                      | 0.61 |  0.36  | 0.01                         | 0.05                      | 0.26                      |  0.12                    |
> | $P^3\text{-}GRPO$ | 8          | Video       | 0.15                  | 0.30                      | 0.54 |**0.33**| 0.00                         | 0.02                      | 0.15                      |**0.06**                   |

---

> > ### Comment · Reviewer_PCAX · 2025-11-27
> >
> > Thank you for the clarification. Your explanation of the architecture design, where Perception-CoT operates on the final frame while Prediction-CoT and Planning-CoT leverage temporal information from the video sequence, is reasonable and well-justified. The ablation study (Image vs. Video) provides clear empirical evidence that temporal information contributes to planning performance. My concern here is adequately addressed.

---

> ### Author Response · Authors · 2025-11-21
> **Thanks and response to Reviewer PCAX (3/4)**
>
> ## Response to Q3: Validation the CoT Dataset
>
> Our CoT dataset is structured at both holistic and modular levels, both critical to the model's performance. **The holistic CoT refers to the label of perception-prediction-planning within the ${P^3\text{-}CoT}$, while the modular CoT ensures the accuracy and consistency of each individual module's reasoning with respect to the video and its labels**.
>
> To guarantee the quality of the ${P^3\text{-}CoT}$ dataset, we adopte a assessment following DriveLM [5]. At the holistic level, each CoT label is manually inspected to ensure it strictly follows the prescribed reasoning structure—progressing completely and sequentially from perception to prediction and then to planning, with all final module outputs present and correctly formatted. At the modular level, we perform a manual check for factual consistency and reasoning quality. The dataset is divided into 10 splits, each assigned to three independent annotators. They randomly sample 10\% of their assigned split and meticulously examine the alignment between the generated CoT and the ground-truth labels for each module. This includes assessing the factual consistency of the reasoning steps, the fluency of the language, and the overall logical plausibility. In both levels of inspection, each reviewed CoT is labeled as "good" or "poor". The accuracy for a split is calculated based on the proportion of "good" samples. We set a passing accuracy threshold of 90\% for a split to be considered qualified. Any split failing to meet this standard is deemed imperfect, and the CoT data for that portion is regenerated and resubmitted for assessment. This iterative process continues until all splits pass the manual quality check by all annotators.
>
> The accuracy and effectiveness of our $P^3\text{-}CoT$ dataset receive strong support from two key experimental results.
>
> **First**, the performance advantage of the full $P^3\text{-}CoT$ over the "only planning" baseline (Table 4, line 5 vs. line 4 of updated manuscript) validates the effectiveness of the holistic CoT structure. Main result as follow:
>
> **Table 4. Only Planning CoT or $P^3\text{-}CoT$**
> | Method | Perception$\uparrow$ | Prediction$\uparrow$ | Planning (Collision,%) 1s$\downarrow$ | Planning (Collision,%) 2s$\downarrow$| Planning (Collision,%) 3s$\downarrow$|Planning (Collision,%) Avg.$\downarrow$ |
> |-|:-:|:-:|:-:|:-:|:-:|:-:|
> | SFT + Only Planning CoT GRPO | -- | -- |0.03|0.08|0.24|0.1|
> | SFT + $P^3\text{-}CoT$ GRPO |**0.64**|**0.54**|**0.00**|**0.02**|**0.15**|**0.06**|
>
> **Second**, the superior performance of the slow-thinking mode (with full CoT) compared to the fast-thinking mode (without reasoning chains) across multiple benchmarks (Tables 1-3 of updated manuscript) confirms the value of accurate and detailed granular CoT reasoning. Together, these results provide comprehensive validation for both the overall framework and the detailed reasoning steps in our $P^3\text{-}CoT$ dataset. We have revised the manuscript and added relevant validation of $P^3\text{-}CoT$ dataset at Appendix B.5. **The main results are the same as in Table 1. (Dual-thinking on nuScenes), Table 2. (Dual-thinking on NAVSIMv1), and Table 3. (Dual-thinking on NAVSIMv2) above**.

---

> > ### Comment · Reviewer_PCAX · 2025-11-27
> >
> > Thank you for detailing the validation process. The two-level inspection methodology (holistic and modular), the multi-annotator setup with a 90% quality threshold, and the iterative regeneration process provide reasonable assurance of dataset quality. The empirical results (Table 4) further support the effectiveness of the P³-CoT structure. I appreciate that this has been added to Appendix B.5.
> > That said, for a core contribution at this venue, the validation methodology could be more rigorous. Specifically:
> > - 10% sampling leaves the majority of the dataset unverified
> > - no inter-annotator agreement metrics (e.g., Cohen's Kappa) are reported
> > - the criteria distinguishing 'good' from 'poor' labels are not explicitly defined.
> > These are not critical flaws, but I would encourage the authors to consider more comprehensive validation protocols in future work or the camera-ready version.
> > Overall, my concern is reasonably addressed, though there is room for improvement.

---

> ### Author Response · Authors · 2025-11-21
> **Thanks and response to Reviewer PCAX (4/4)**
>
> ## Response to Q4: CoT or Model?
>
> On the one hand, as shown in Table 4 of updated manuscript, ${P^3\text{-}CoT}$ significantly reduces the collision rate compared to the "only planning" baseline **(see Table 1.4 (Only Planning CoT or $P^3\text{-}CoT$))**. Furthermore, our "dual-thinking" ablation study provides additional validation: the "detailed-thinking" mode, which produces complete reasoning chains, consistently outperforms the "fast-thinking" mode that outputs only final answers across nuScenes and NVASIMv1/v2 benchmarks (Tables 1-3 of updated manuscript). These results collectively demonstrate that the performance gain stems directly from the CoT data itself, instead of foundation model training, as all other training settings remain identical across these ablation studies. **The main results are the same as in Table 1.1 (Dual-thinking on nuScenes), Table 1.2 (Dual-thinking on NAVSIMv1), and Table 1.3 (Dual-thinking on NAVSIMv2) above**.
>
> On the other hand, as shown in Table 1 of updated manuscript, for the methods using **the same VLM (Qwen2.5-VL-3B)** as ours, such as AutoVLA and OpenDriveVLA, our model trained on ${P^3\text{-}CoT}$ with ${P^3\text{-}GRPO}$ achieves better performance, especially in collision rate, which proves that the improvement of our model is gained from our CoT and GRPO; for the methods using **the larger VLM**, such as OmniDrive using LLava-7B and DriveVLM using Qwen2-VL-7B, our model still gains better performance with a smaller VLM.  These results collectively demonstrate the effectiveness of CoT, instead of the higher capacity of the model generally. Main result as follow:
>
> | Method                                    | L2(m) 1s$\downarrow$ | L2(m) 2s$\downarrow$ | L2(m) 3s$\downarrow$ | L2 (m) Avg.$\downarrow$ | Collision(%) 1s$\downarrow$ | Collision(%) 2s$\downarrow$ | Collision(%) 3s$\downarrow$  | Collision(%) Avg.$\downarrow$ | VLM           |
> |-------------------------------------------|:-----------:|:-----------:|:-----------:|:-------------:|:-----------------:|:-----------------:|:------------------:|:-------------------:|:---------------:|
> | DriveVLM (24' CoRL)        | 0.18      | 0.34      | 0.68      | 0.40        | 0.10            | 0.22            | 0.45             | 0.27              | Qwen2-VL-7B   |
> | OmniDrive (25' CVPR)     | 0.14      | 0.29      | 0.55      | 0.33        | 0.01            | 0.04            | 0.27             | 0.11              | LLava-7B      |
> | OpenDriveVLA (26' AAAI)| 0.14      | 0.30      | 0.55      | 0.33        | 0.02            | 0.07            | 0.22             | 0.10              | Qwen2.5-VL-3B |
> | AutoVLA (25' NeurIPS)          | 0.25      | 0.46      | 0.73      | 0.48        | 0.07            | 0.07            | 0.26             | 0.13              | Qwen2.5-VL-3B |
> | $AutoDrive\text{-}P^3$ (Ours-Detailed)    |**0.15**|**0.30**|**0.54**|**0.33**|**0.00**|**0.02**|**0.15**|**0.06**| Qwen2.5-VL-3B |
> | $AutoDrive\text{-}P^3$ (Ours-Fast)        | 0.16      | 0.31      | 0.56      | 0.34        | 0.00            | 0.04            | 0.20             | 0.08              | Qwen2.5-VL-3B |

---

> > ### Comment · Reviewer_PCAX · 2025-11-27
> >
> > Thank you for the comprehensive ablation analysis. The combination of (1) controlled internal comparisons (Table 4, dual-thinking), (2) same-VLM baselines (AutoVLA, OpenDriveVLA), and (3) outperforming larger models (DriveVLM, OmniDrive) provides convincing evidence that the improvements stem from the P³-CoT framework rather than model capacity alone. My concern is adequately addressed.

---

> ### Author Response · Authors · 2025-11-28
> **Official Comment by Authors**
>
> Dear Reviewer PCAX,
>
> Thank you for your timely and constructive feedback. We are pleased to hear that our responses have addressed most of your concerns and raising score. Following your suggestion, we have revised our claim from **"VLM-based AD needs CoT"** to **"VLM-based systems benefit from CoT"** in our updated manuscript. We truly appreciate your insightful comments, which have been invaluable in improving our work.
>
> Best regards,
>
> The Authors

---

### Author Response · Authors · 2025-12-04
**Summary of the Score Changes and Discussions in Rebuttal**

Dear Area Chairs,

For the convenience of your review, we have made the following summary for rebuttal.

This paper present ${AutoDrive\text{-}P^3}$, resolving a key limitation of current VLMs by explicitly capturing the relationship between perception, prediction, and planning in autonomous driving. On open-loop benchmark (nuScenes), we achieve the same level as SOTA methods at L2 and overpass about 40\% compared to SOTA methods at collision rare. On closed-loop (NAVSIMv1/v2) benchmarks, we achieve SOTA planning metrics (90.6 PDMS and 89.9 EPDMS). Additionally, to balance inference efficiency with performance, we introduce dual thinking modes: detailed thinking and fast thinking.

Before the system freezed, our submission had effectively increased from scores **8,6,6,4 (confidence 4,4,3,5)** to scores **8,8,6,4 (confidence 4,5,3,5)**, with one reviewer (**PCAX**) explicitly confirming score increases.

The other reviewers do not have a chance to reply, but we provide explanations and supplementary experiments for each of their questions, including validity of CoT concerns of Reviewers NDzW & DPeP. And we also appended dual system and running analysis to address the effectiveness and efficiency concerns of Reviewer K3hP. Specifically, as shown below:

1. Reviewer **NDzW** is concerned about how CoT works and insufficient reference of VLM-based methods. Furthermore, Reviewer NDzW asks for more details about the detailed prompt and the validation method of CoT. For how CoT works, we give further explanation of our visualization examples and ablation experiment results to prove CoT effectiveness (`Line 253-258 and Table 1-4 of updated manuscript`). For reference, we add additional reference. Additionally, we discuss the limitation of L2 distance and add extra experiments on NAVSIMv2 to show the effectiveness of our methods (`Table 3 of updated manuscript `). For prompts and validation method of CoT, we give comprehensive introduction in appendix (`Appendix B.5 and Table 1-3 of updated manuscript`).

2. Reviewer **DPeP** is concerned about the difference of CoT generated by different generative models and validation of our CoT. The error accumulation of hierarchical supervision, reward weight setting and the benefits of hierarchical supervision are also mentioned. For the first question, we claim that there is almost no difference between different generative models and validation methods to prove the effectiveness of CoT (`Line 253-258 and Table 1-4 of updated manuscript`). For error accumulation, we further discuss about hierarchical supervision of ${P^3\text{-}CoT}$. For reward weight setting, we provide ablation study under different settings (`Line 1052, Table 10 of updated manuscript`). For the benefits of hierarchical supervision, we further explain our ablation experiments to show the benefits of our methods (`Line 432, Table 4 of updated manuscript`).

3. Reviewer **K3hP** is concerned about the runtime and hybrid setting of our systems. For runtime and hybrid setting, we add dual-thinking experiments. The results show that our fast-thinking mode maintains competitive performance and reach real-timing requirements (`Table 1-3, Figure 5 of updated manuscript`).

We sincerely thank you for the additional time and effort devoted to reviewing and evaluating our submission and hope these clearly comments and score changes will be taken into consideration when making the final recommendation.

Best regards,

The Authors

---

### Meta-Review · Area_Chair_SZ6r · 2025-12-12

**Summary:**

Advantages:
1. Proposed AutoDrive-P³, integrating CoT reasoning on perception, prediction, and planning within a VLM, together with P³-CoT dataset, compared to other planning-only approaches.
2. A hierarchical reinforcement learning algorithm P³-GRPO was presented for progressive supervision across all three tasks.
3. Experiments on both open-loop nuScenes and closed-loop NavSim showed competitive results with increasing interpretability.

Weaknesses:
1. The quality and correctness of the P³-CoT dataset generated by Qwen2.5-VL-72B was hard to be justified.
2. The algorithm was complex, with long computation time. Besides, the rebuttal baselines chosen with outdated methods were not convinced.
3. Clear videos should be offered to show how the algorithm worked better than others and why, while the score was just a scalar.

**Reviewer Concerns:**

Reviewer PCAX (Johannes Betz) was addressed.
Part of Reviewer K3hP (Zaiwei Zhang) was not.

**Reviewer Scores:**

Reviewer PCAX (Johannes Betz) replied to increase score. Others did not reply, and I do not think they will change.

---

### Decision · Program_Chairs · 2026-01-26

Accept (Poster)